# Upper critical solution temperature polymer assemblies via variable temperature liquid phase transmission electron microscopy and liquid resonant soft X-ray scattering

Joanna Korpanty [1], Cheng Wang [2] & Nathan C. Gianneschi [1,3,4] ✉

Here, we study the upper critical solution temperature triggered phase transition of thermally responsive poly(ethylene glycol)-*block*-poly(ethylene glycol) methyl ether acrylate-*co*-poly(ethylene glycol) phenyl ether acrylate-*block*-polystyrene nanoassemblies in isopropanol. To gain mechanistic insight into the organic solution-phase dynamics of the upper critical solution temperature polymer, we leverage variable temperature liquid-cell transmission electron microscopy correlated with variable temperature liquid resonant soft X-ray scattering. Heating above the upper critical solution temperature triggers a reduction in particle size and a morphological transition from a spherical core shell particle with a complex, multiphase core to a micelle with a uniform core and Gaussian polymer chains attached to the surface. These correlated solution phase methods, coupled with mass spectral validation and modeling, provide unique insight into these thermoresponsive materials. Moreover, we detail a generalizable workflow for studying complex, solution-phase nanomaterials via correlative methods.

For upper critical solution temperature (UCST) homopolymers, heating above the transition temperature results in an enthalpically driven solubilization, as interactions between polymer chains are weakened in favor of polymer-solvent interactions[1–3]. In multiblock copolymer amphiphiles, the inclusion of a UCST block can enable temperature triggered nanoscale morphological transitions. The ability of UCST polymers to undergo thermally induced phase transitions holds promise for numerous applications, ranging from insulating materials to catalysis[4–6]. Despite their utility, there are few studies concerned with probing UCST transitions as these types of phase transformations cannot be imaged by standard microscopy methods that rely on static imaging of dried aliquots. Moreover, the majority of UCST-type polymers do not exhibit thermoresponsiveness under purely aqueous conditions[3,7–11]. This limits the utility of cryogenic TEM in deciphering morphological transitions because, despite there being limited examples of extending cryo-TEM to non-aqueous solvents[12–14], standard methods are optimized for aqueous samples only[12,15,16]. Moreover, even applying scattering techniques, such as liquid-phase X-ray scattering, to characterize UCST materials presents a challenge because appropriate model selection for data fitting is nontrivial and is often guided by TEM data[17–20].

To elucidate the morphology and dynamics of UCST polymeric nanomaterials in organic solvents, we propose a comparative approach coupling variable temperature liquid-cell transmission electron microscopy (VT-LCTEM) and variable temperature liquid resonant soft X-ray scattering (VT-RSoXS). VT-LCTEM is a powerful solution-phase nanoscale imaging technique with in situ heating capabilities where two silicon microchips fabricated with electron

[1]Department of Chemistry, International Institute for Nanotechnology, Chemistry of Life Processes Institute, Simpson Querrey Institute, Northwestern University, Evanston, IL 60208, USA. [2]Advanced Light Source, Lawrence Berkeley National Laboratory, 1 Cyclotron Road, Berkeley, CA 94720, USA. [3]Department of Materials Science & Engineering, Northwestern University, Evanston, IL 60208, USA. [4]Department of Biomedical Engineering and Department of Pharmacology, Northwestern University, Evanston, IL 60208, USA. ✉e-mail: Nathan.gianneschi@northwestern.edu

transparent silicon nitride (SiN$_x$) windows are used to hermetically seal a liquid sample (Fig. 1)[21,22]. In turn, VT-RSoXS can utilize the same sample geometry and the same type of heated liquid-cell (Fig. 1a)[23–28]. This shared liquid-cell design allows for the same experimental conditions to be examined by both measurements, providing real space and reciprocal space insight into solution phase nanomaterials. Critically, our study points to the value of leveraging LCTEM and RSoXS in tandem to study materials with complex, responsive behavior in solution.

For the correlative investigation of UCST materials at variable temperature, we designed a proof-of-concept UCST-type polymer capable of forming responsive nanoassemblies in an organic solvent, isopropanol (IPA). We chose the triblock copolymer poly(ethylene glycol)-*block*-poly(ethylene glycol) methyl ether acrylate-*co*-poly(ethylene glycol) phenyl ether acrylate-*block*-polystyrene (PEG-*b*-PEG-MeA-*co*-PEGPhA-*b*-PS), which was synthesized via reversible addition fragmentation chain transfer (RAFT) polymerization induced self-assembly (PISA) in IPA (Fig. 1). For PEG-*b*-PEGMeA-*co*-PEGPhA-*b*-PS, the PEG block serves as the stabilizer block, the PS block serves as the solvophobic block, and the PEGMeA-*co*-PEGPhA block serves as the UCST block; the amphiphilicity and inclusion of a UCST block in the polymer enables the formation of responsive nanoassemblies.

## Results and discussion

PEG-*b*-PEGMeA-*co*-PEGPhA-*b*-PS was synthesized via sequential RAFT polymerization in IPA (Fig. 1b). The installation of the PS block, and thus the formation of nanoassemblies, was enabled via a final thermal PISA step. In accordance with PISA requirements, holding the macro chain transfer agent PEG-*b*-PEGMeA-*co*-PEGPhA above the UCST made the entire polymer soluble, as confirmed by variable temperature dynamic light scattering (VT-DLS, Supplementary Fig. 1). As PISA has neither been applied to such a UCST-type polymer nor been per-

formed in pure IPA[29–31], we first confirmed successful polymerization via size exclusion chromatography multiangle light scattering (SEC-MALS), and we verified that nanoassemblies indeed formed via DLS (Fig. 1b–d). To probe the thermal response of the PEG-*b*-PEGMeA-*co*-PEGPhA-*b*-PS triblock copolymer, we subjected the assemblies to variable temperature (VT) DLS, where we observed that heating the assemblies prompted a decrease in diameter upon solubilization of the responsive PEGMeA-*co*-PEGPhA block (Fig. 1d). Additionally, by VT-DLS, the distribution of the bulk assemblies had become less polydisperse above the UCST, with a polydispersity index (PDI) of 0.22 before heating and 0.13 at 60 °C. The initial size was not restored upon cooling, leaving a less polydisperse distribution with a PDI of 0.07. Thus, though we obtain limited insight through indirect bulk scattering, VT-DLS reveals that annealing the assemblies ultimately yields more ordered nanostructures.

Next, to directly characterize the initial morphology of the polymer, we employed dry state TEM, where we observed somewhat polydisperse, aggregated spheres (Fig. 1e). Given that the thermal PISA process used to prepare the PEG-*b*-PEGMeA-*co*-PEGPhA-*b*-PS polymer is quenched by cooling the hot polymer solution from above to below the UCST of PEGMeA-*co*-PEGPhA, the morphological polydispersity is unsurprising, with the cooling process resulting in kinetic trapping of the polymer assemblies[32–34]. As PS has a high glass transition temperature (~100 °C), we also synthesized and investigated PEG-*b*-PEG-MeA-*co*-PEGPhA-*b*-P(n-butyl acrylate), made with a lower glass transition temperature core-forming block, poly(n-butyl acrylate) (PnBuA, T$_g$ – 50 °C), Supplementary Fig. 2. Prepared via the same PISA process, PEG-*b*-PEGMeA-*co*-PEGPhA-*b*-PnBA showed prohibitively low contrast in the dry state and appeared virtually invisible in liquid IPA by TEM, preventing further investigation (Supplementary Fig. 2). Accordingly, we pursued further studies with the PEG-*b*-PEGMeA-*co*-PEGPhA-*b*-PS UCST polymer, as few other compatible core-forming

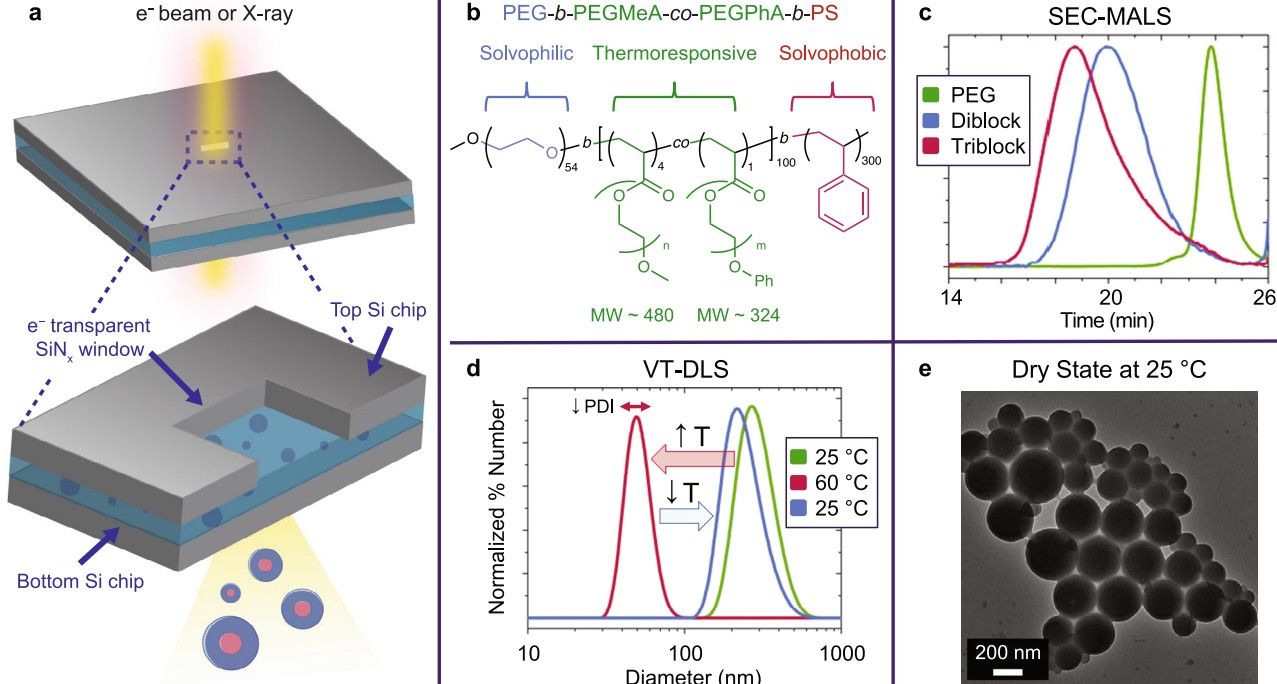

**Fig. 1 | Liquid-cell sample geometry and the UCST polymer, poly(ethylene glycol)-*block*-poly(ethylene glycol) methyl ether acrylate-*co*-poly(ethylene glycol) phenyl ether acrylate-*block*-polystyrene (PEG-*b*-PEGMeA-*co*-PEGPhA-*b*-PS). a** The liquid-cell platform is used for both LCTEM and liquid RSoXS. **b** Structure of PEG-*b*-PEGMeA-*co*-PEGPhA-*b*-PS. **c** Size exclusion chromatography multiangle light scattering (SEC-MALS) traces for PEG, PEG-*b*-PEGMeA-*co*-PEGPhA, and PEG-*b*-PEGMeA-*co*-PEGPhA-*b*-PS. **d** Variable temperature dynamic light scattering (VT-DLS) of UCST polymer solution in IPA. **e** Dry state TEM of PEG-*b*-PEGMeA-*co*-PEG-PhA-*b*-PS dried from IPA.

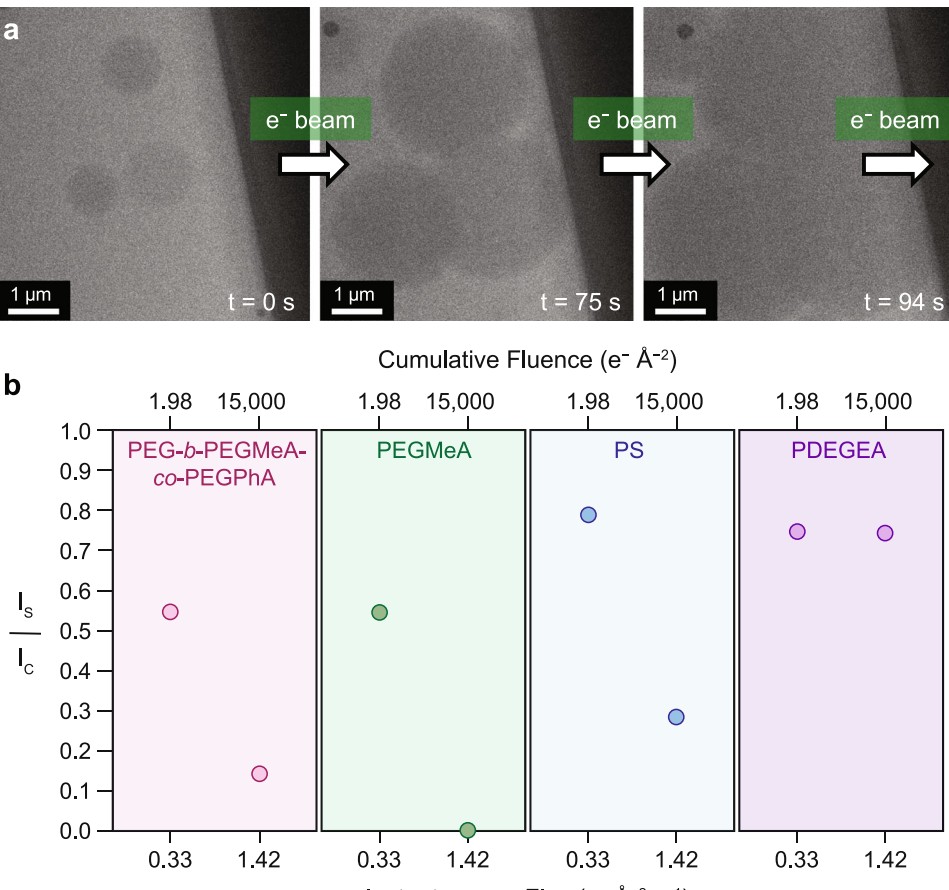

**Fig. 2 | Beam damage diagrams developed from LCTEM and MALDI-IMS analysis of PEG-*b*-PEGMeA-*co*-PEGPhA-*b*-PS triblock copolymer nanoassemblies.** **a** LCTEM experiment on PEG-*b*-PEGMeA-*co*-PEGPhA-*b*-PS continuously imaged at a flux of 1.42 e⁻ Å⁻² s⁻¹. **b** MALDI-IMS damage plot where $I_s/I_c$ is measured under different imaging conditions for different polymer damage experiments. Here, $I_s/I_c$ denotes the ratio of imaged sample mass signal intensity ($I_s$) over unimaged control mass signal intensity ($I_c$). The ratio $I_s/I_c$ was plotted as a function of instantaneous flux and cumulative fluence. Three polymers were investigated for these MALDI-IMS damage studies: PEG-*b*-PEGMeA-*co*-PEGPhA, PDEGMeA, PDEGEA, and PS.

blocks fit the strict requirement of PISA for a soluble monomer that yields an insoluble block in IPA.

Prior to LCTEM analysis of the responsive assemblies, we endeavored to establish safe imaging conditions to understand electron beam induced effects[20,35–37]. First, a kinetic model for the radiolysis[38,39] of IPA under LCTEM conditions was built and used to compare the stability of IPA to water, the most common solvent used for LCTEM studies (Supplementary Fig. 3, Supplementary Tables 1, 2)[40–50]. In brief, we found that IPA forms lower reactivity radicals at lower steady state concentrations compared to water, suggesting that IPA is more amenable to LCTEM studies than water. Moreover, we reasoned, the lower density of IPA compared to water should allow for improved visibility of low contrast, solution-phase organic materials (see SI for discussion)[38].

To experimentally characterize beam damage effects in the absence of heating, we subjected the solution-phase assemblies to continuous LCTEM imaging at a flux of 1.42 e⁻ Å⁻² s⁻¹ to a cumulative fluence of 15,000 e⁻ Å⁻² (Fig. 2). These high flux continuous imaging experiments led to beam induced film formation over the entire imaged region (Fig. 2a, Supplementary Movie 1). Upon completion of LCTEM damage experiments, we studied the extent of polymer degradation via a separate matrix-assisted laser desorption/ionization imaging mass spectrometry (MALDI-IMS) experiment[20,38]. Here, MALDI-IMS was used to evaluate the mass signal intensity of the polymer distribution over the imaged area. Because the triblock copolymer does not efficiently ionize[20], we performed these LCTEM

and MALDI-IMS damage experiments on lower molecular weight proxies for the UCST triblock copolymer (Supplementary Figs. 4–6). Specifically, we subjected polymers representative of each block, PEG-*b*-PEGMeA-*co*-PEGPhA, PEGMeA, and PS, to two LCTEM experiments each, using low (0.33 e⁻ Å⁻² s⁻¹, <2 e⁻ Å⁻²) and high (1.42 e⁻ Å⁻² s⁻¹, 15000 e⁻ Å⁻²) flux conditions (Supplementary Fig. 6, Supplementary Movies 2-5). The low flux condition of 0.33 e⁻ Å⁻² s⁻¹ was chosen because it enabled visibility of nanostructures without clear structure degradation during imaging. We note that given the complex structure and thus complex fragmentation of PEG-*b*-PEGMeA-*co*-PEGPhA and PEGMeA, we also synthesized poly(di(ethylene glycol) ethyl ether acrylate) (PDEGEA) to evaluate the survival of the polymer backbone, as the mass signals for the two former polymers are dominated by oligo ethylene glycol (OEG) sidechains, leading to a peak spacing of 44 m/z (Supplementary Fig. 5). Data from these experiments were used to develop beam damage diagrams where the ratio of the mass signal intensity for different imaged samples ($I_s$) to the intensity of an unimaged control ($I_c$) was plotted as a function of the instantaneous flux and cumulative fluence (Fig. 2b). The unimaged control sample was prepared by dropcasting the same polymer solution used in LCTEM damage experiments onto a liquid-cell chip and then immediately conducting MALDI-IMS. Here, the ratio $I_s/I_c$ was used to evaluate sample survival under different LCTEM imaging conditions, with a value of $I_s/I_c$ approaching 1 indicating less polymer degradation. As revealed by MALDI-IMS, all of the examined polymers showed reduced $I_s/I_c$ values under high flux conditions, indicating enhanced sample

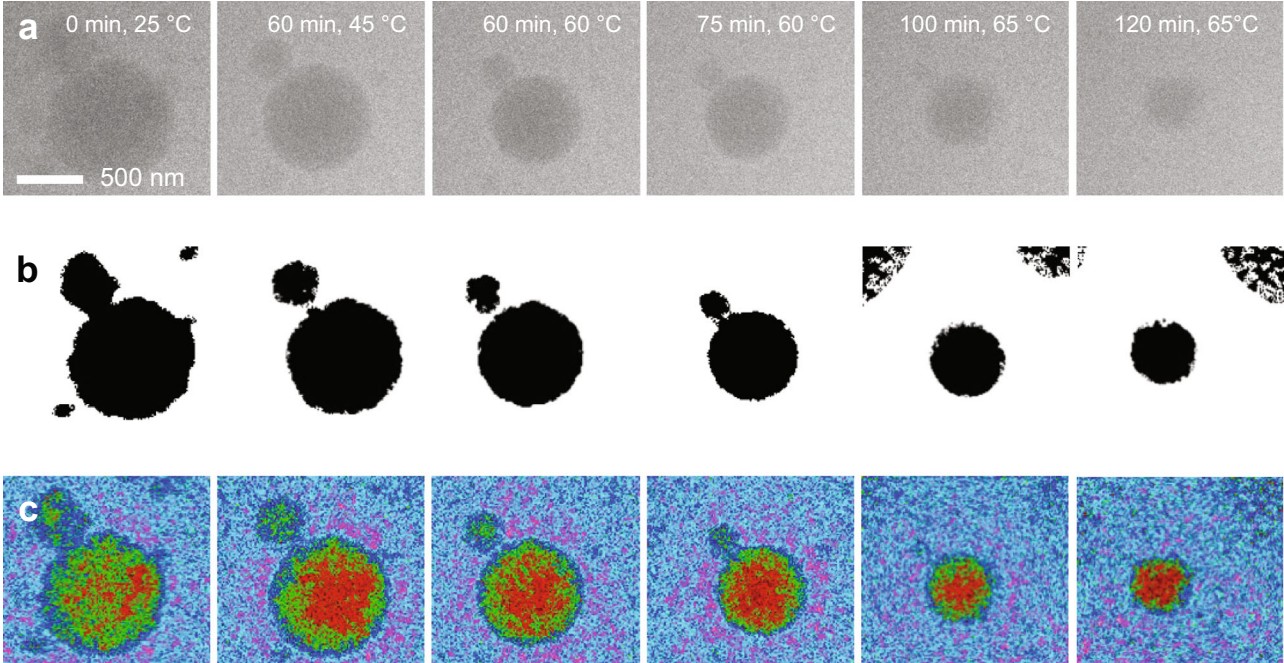

**Fig. 3 | VT-LCTEM experiment on PEG-*b*-PEGMeA-*co*-PEGPhA-*b*-PS in IPA imaged at a flux of 0.33 e⁻ Å⁻² s⁻¹ and heated to 65 °C. a** Region of interest in raw VT-LCTEM data over course of heating experiment (see Supplementary Fig. 7 for uncropped images). **b** Segmented region of interest for each timepoint. **c** False colored region of interest for each timepoint shown with a 6-shade false color filter.

degradation (Fig. 2b, Supplementary Figs. 4–6, see SI for more discussion)[51]. On the other hand, low flux conditions showed enhanced mass signal retention, allowing us to limit the degradation of the examined polymers.

With gentle imaging conditions established, we imaged the UCST polymeric assemblies via VT-LCTEM at a flux of 0.33 e⁻ Å⁻² s⁻¹ (Fig. 3). Here, we employed stroboscopic rather than continuous imaging conditions to limit the extent of electron beam induced damage to the polymeric materials[20,35]. In agreement with the VT-DLS data, upon heating to 65 °C, the PEG-*b*-PEGMeA-*co*-PEGPhA-*b*-PS polymer assemblies underwent contraction, a phenomenon that was reproduced over several experiments (Fig. 3a, Supplementary Figs. 7–9). The distinction between the behavior observed under low flux, stroboscopic imaging conditions and the higher flux, continuous imaging conditions suggests that the UCST-triggered particle size reduction is likely not beam-induced (Figs. 2a, 3a). Moreover, under low flux stroboscopic conditions with no heating, the nanoassemblies did not change in size, further suggesting that the UCST transition is indeed responsible for the observed particle contraction (Supplementary Fig. 10). The film formation process, on the other hand, observed under high flux conditions is attributed to damage, as corroborated by MALDI-IMS (Fig. 2a). Thus, our results suggest that under low flux conditions, the observed particle behavior is likely the result of the UCST transition and not the result of electron beam damage.

To aid in image interpretation of the UCST transition observed under low flux conditions, we performed image processing on a cropped region of interest (ROI) using Fiji (Fig. 3b, c)[20,52]. Thresholding and segmenting the ROI highlights the sequential reduction in particle diameter with continued heating, as expected for the UCST transition (Fig. 3b). Rather than proceeding via fission, the reduction in particle size proceeds purely through a reduction in particle diameter, ultimately resulting in smaller spherical assemblies of lower morphological polydispersity, as confirmed by *post mortem* dry state TEM of the dried LC chip (Supplementary Fig. 11). To gain insight into the UCST transition, we applied a background subtraction, 4 ×4 average binning,

and a false color filter ("6-shade" lookup table) by which each gray scale pixel was assigned a corresponding RGB value (Fig. 3c). This image processing procedure allows us to probe subtle differences in density and highlights that initially, the core is heterogenous, presumably containing mixed regions of PS and PEGMeA-*co*-PEGPhA surrounded by a PEG shell (Figs. 2a, 3c). Upon heating, the PEGMeA-*co*-PEGPhA regions were expelled from the now PS-rich core, and these PEGMeA-*co*-PEGPhA-rich regions became less visible with further heating. Overall, these false colored images suggest, upon the UCST transition, the thermoresponsive PEGMeA-*co*-PEGPhA block disentangles from the core, leaving a PS-rich core and becomes solubilized in the surrounding solvent. We note that the observed transformation does not appear to be reversible via VT-LCTEM, which may be because of confinement combined with the entropic barrier of the reverse transformation, a feature we aimed to explore through correlative analysis in similarly confined liquid RSoXS experiments (Supplementary Fig. 12).

In recent work, to complement insight into transformations directly observed by VT-LCTEM, VT-SAXS was utilized[20]. While VT-SAXS is a powerful, bulk technique for characterizing thermoresponsive nanomaterials, SAXS scattering contrast relies on differences in electron density between the sample and solvent at high photon energies (~2500 eV)[53]. The use of hard X-rays in SAXS provides limited chemical insight into scattering objects and requires that samples be measured in capillaries, potentially posing difficulties in correlating real space LCTEM data with X-ray scattering data due to their distinct sample geometries[26,27]. Unlike SAXS, small angle neutron scattering (SANS) can enable contrast-matching to selectively analyze scattering from certain parts of a given nanostructure. However, to achieve contrast-matching, the sample must be modified via radiolabeling, requiring additional synthesis that may not be straightforward for every system[54]. Moreover, the low flux of neutron sources compared to X-ray sources requires longer measurement times to obtain sufficient statistics[55]. Like SAXS, SANS also differs greatly in sample geometry from LCTEM, potentially posing difficulties in correlating the two measurements.

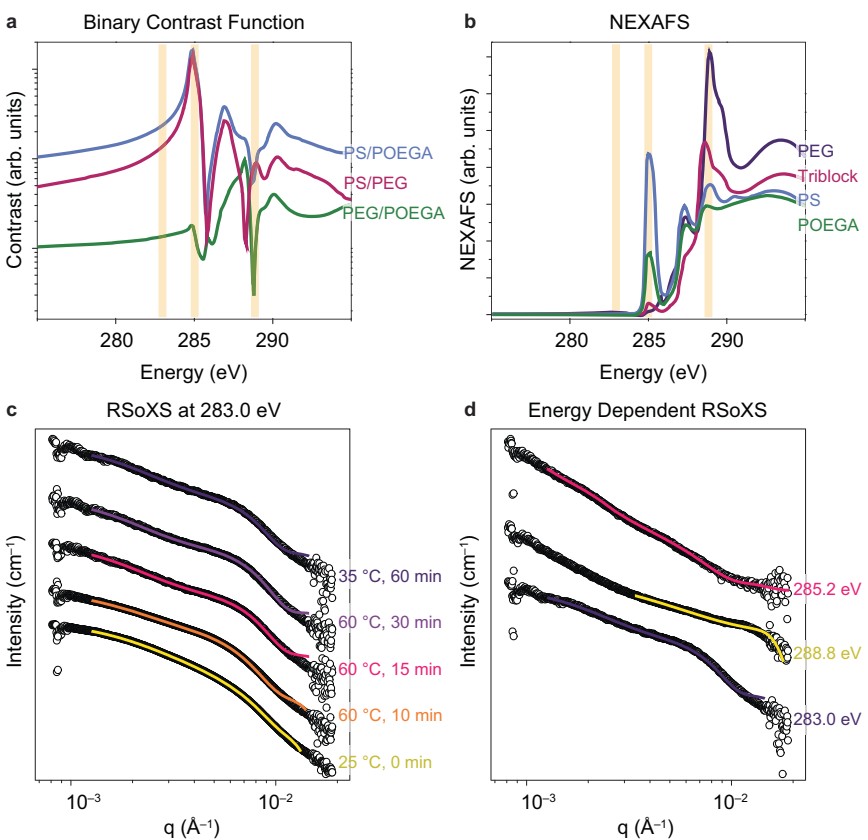

**Fig. 4 | X-ray scattering data measured for a solution of PEG-*b*-PEGMeA-*co*-PEGPhA and PEGMeA in IPA. a** Binary contrast functions between different homopolymers, where POEGA is an abbreviation for the oligoethylene glycol copolymer PEGMeA-*co*-PEGPhA. **b** NEXAFS measurements for homopolymers and UCST triblock copolymer. **c** RSoXS data measured at a fixed energy of 283.0 eV. **d** RSoXS data measured upon cooling sample and holding at room temperature for 30 minutes using energies of 285.2 eV, 288.8 eV, and 283.0 eV.

Given the limitations of SAXS and SANS, we turned here to liquid resonant soft X-ray scattering (RSoXS), where the large scattering cross section of soft X-rays enables analysis of thin transmission samples used in TEM (Fig. 4)[24,25,27,56]. Liquid RSoXS is capable of probing the structure of complex, multicomponent polymeric nanoarchitectures because of its ability to achieve contrast matched scattering through variation of incident X-ray energy[57]. In RSoXS, the scattering intensity is proportional to the energy-dependent contrast function $I\alpha|\Delta n(E)|^2$, where $\Delta n$ denotes the difference between the indices of refraction for two chemical moieties and is parameterized by real and imaginary components via the relation $n(E) = 1-\delta(E) + i\beta (E)$[28]. The real and imaginary components are typically determined via near-edge X-ray absorption fine structure (NEXAFS) spectroscopy. Guided by the contrast function, energies near the carbon K-edge can be selected for RSoXS measurements. For example, selecting an energy that allows for contrast matching of the corona block to the solvent enables isolation of scattering contributions from the core[23]. Fitting these scattering data to established models can then provide critical insight into the size and shape of the core. Moreover, by using time-resolved in situ RSoXS, we can monitor the morphological development of nanostructures through changes in the form factor. RSoXS data can then be correlated to phenomena directly observed by LCTEM imaging, allowing us to analyze complex phase transformations that would otherwise be challenging to monitor by either RSoXS or LCTEM in isolation. Critically, both LCTEM and liquid RSoXS employ the same liquid-cell holder tip assembly, and thus both experiments subject the polymeric sample to the same inherent confinement effects (Supplementary Fig. 12).

Prior to conducting RSoXS measurements, we performed NEXAFS spectroscopy measurements on pure PS, PEGMeA-*co*-PEGPhA, and PEG films to generate binary contrast functions for the polymers (Fig. 4a, b). At an energy of 283 eV, the contrast between PEGMeA-*co*-PEGPhA and PEG is small, while the contrast between PS and these two polymers is large. Thus, we used an energy of 283 eV to probe the overall core-shell(-shell) morphology of the PEG-*b*-PEGMeA-*co*-PEG-PhA-*b*-PS polymer structures, as this energy showed the most significant scattering intensity for energies near the carbon K-edge (Fig. 4c, Supplementary Fig. 13)[27]. Using an energy of 283 eV, at room temperature, a broad trace was observed, indicative of the high polydispersity of the initial assemblies before heating. The room temperature data was fit to a spherical polymer micelle form factor, a morphological model informed by liquid-phase real space imaging. Using this fit suggested that, on average, the initial assemblies had a core radius of 47 nm with a coronal thickness of 48 nm. Upon heating the sample to 60 °C for 10 minutes, a scattering feature was observed at q ̴ 0.007 nm⁻¹, suggesting ordering of the nanoassemblies, in agreement with VT-DLS data showing a tightening of the distribution. Fitting this trace to a polymer micelle form factor showed an approximate core radius of 42 nm and coronal radius of 59 nm. The decrease in the size of the core can be explained by the enhanced solubility of the PEGMeA-*co*-PEGPhA block upon the UCST transition. Presumably, at room temperature, the core is PS and PEGMeA-*co*-PEGPhA-rich, as PEG is the only solvophilic block at room temperature and should thus be found in the outer shell. However, upon heating, PEGMeA-*co*-PEGPhA becomes increasingly solvophilic, leading to an effective increase of the solvophilic corona and a decrease in the insoluble core. This outer corona is "invisible" by LCTEM since this shell is entirely soluble in isopropanol (Fig. 3c). After being held for 5 additional minutes at 60 °C, we observed a

PEG-*b*-PEGMeA-*co*-PEGPhA-*b*-PS

**Fig. 5 | Mechanism for UCST transformation in PEG-*b*-PEGMeA-*co*-PEGPhA-*b*-PS.** Blue: polyethylene glycol (PEG). Green: poly(ethylene glycol) methyl ether acrylate-*co*-poly(ethylene glycol) phenyl ether acrylate (PEGMeA-*co*-PEGPha). Red: polystyrene (PS).

discernible scattering feature. We applied a polymer micelle form factor, yielding a fit with a core radius of 39 nm with coronal thickness of 73 nm, suggesting further expulsion of PEGMeA-*co*-PEGPhA from the particle core, consistent with TEM imaging data (Supplementary Fig. 10). By RSoXS, the nanoassemblies remained in this morphology after being held at 60 °C for 15 more minutes, with a core radius of 38 nm and coronal radius of 75 nm. Upon cooling and holding the sample at <35 °C for 30 minutes, the sample was fit to a polymer micelle form factor with a core radius of 38 nm with coronal thickness of 65 nm, suggesting partial reversibility of the UCST transformation given the decrease in coronal thickness.

To further probe the final, trapped morphology after the heating-cooling cycle, we conducted energy dependent RSoXS at 283 eV, 285.2 eV, and 288.8 eV (Fig. 4d). While the 283 eV data describes the overall micellar morphology, using an energy of 285.2 eV allowed us to enhance contrast from the PS domains. Using a spherical form factor with a sticky hard sphere structure factor, the fit for the 285.2 eV trace revealed a 21 nm PS core radius. As the 283 eV fit shows that the entire core radius was 48 nm, the smaller radius measured at 285.2 eV suggests that, upon cooling, the core likely contains both PS and PEGMeA-*co*-PEGPhA, pointing to a partial reversibility of the UCST transition.

Using an energy of 288.8 eV results in similar contrast for all three polymer domains, allowing us to probe the overall particle size[24,25]. Given the similarity in contrast for all three domains, the 288.8 eV trace showed low scattering contrast, and so the overall scattering can be more easily observed by plotting ln ($I \times Q^2$) vs ln Q plot (Supplementary Fig. 14). Additionally, the low scattering contrast may partially be attributed to our inability to perform a solvent background subtraction due to the variability of liquid thickness over separate experiments. In future studies, this issue may be ameliorated by creating a flow cell capable of flowing preformed nanomaterials for liquid RSoXS measurements. Nonetheless, fitting the 288.8 eV trace to a spherical form factor with a sticky hard sphere structure factor showed an overall sphere diameter of 130 nm. Due to our use of simplified models, we stress the importance of the ability of RSoXS to probe the dynamic behavior of the sample and relative change in size upon heating, rather than the precise size of scattering features. Despite our simplified approach, it is worth noting that for the energies that we have investigated near the carbon K-edge, the isopropanol solvent itself has a strong cross-section. This consideration has likely prevented others from pursuing liquid RSoXS studies in organic solvents, making our study the first of its kind. We also note that IPA likely has a complicated index of refraction, which is challenging to measure and so in our analysis, we are missing an optical parameter and set of contrast functions associated with the solvent. However, even with these limitations and without background subtraction, we were still able to obtain sufficient signal intensity to probe the UCST transition, demonstrating the utility of liquid RSoXS to investigate non-aqueous systems in future studies.

Correlating liquid RSoXS data with VT-LCTEM imaging results, we propose an overall mechanism for the UCST triggered transformation. Specifically, by both techniques, the UCST appears to prompt a transition from a PEGMeA-*co*-PEGPhA and PS-rich core to a PS-rich core. This core shrinking is accompanied by coronal swelling, as PEGMeA-*co*-PEGPhA becomes solubilized by IPA and joins PEG in the particle corona (Fig. 5). Upon cooling, the assemblies show partial reversibility of the UCST transformation, likely due to the high glass transition temperature of the PS core. As demonstrated for this UCST polymer system, leveraging LCTEM and RSoXS in tandem provides a general approach that can be used to obtain valuable insight into the morphology and stimuli-responsiveness of complex, solution-phase nanomaterials.

In summary, we have gained insight into complex polymeric nanostructures and their dynamics by studying a UCST polymer, PEG-*b*-PEGMeA-*co*-PEGPhA-*b*-PS, in isopropanol after establishing safe imaging conditions. Critically, we correlate VT-LCTEM data with liquid VT-RSoXS, subjecting the sample to the same experimental conditions for both measurements. By leveraging the techniques in tandem, we obtained key insights into the complex structure and dynamics of PEG-*b*-PEGMeA-*co*-PEGPhA-*b*-PS, notably observing a UCST-triggered change in morphology from a polydisperse core-shell structure with a complex core to a more-ordered spherical micelle. Our generalizable workflow shows the potential of LCTEM, coupled with X-ray scattering, to answer fundamental questions about functional and responsive nanomaterials. We propose this approach can be extended to study complex nanoscale processes in both aqueous and organic-solvated systems.

## Methods

### General Information

All materials were purchased from Sigma or TCI chemicals. All monomers were filtered through basic alumina to remove inhibitors and all other materials were used as received unless otherwise noted. The synthetic details for polymeric materials are provided in the supplementary information.

### LCTEM imaging

A Protochips Poseidon Select Heating holder was used to collect LCTEM data. Isopropanol was used to prefill the lines of the holder in all LCTEM experiments. LCTEM chips with 50-nm-thick, 200 µm × 50 µm window SiN$_x$ membranes were cleaned in acetone followed by methanol, dried, and subsequently glow discharged in a PELCO easiGlow glow discharge unit for 5 min. Next, 0.5 µL of sample was pipetted manually onto the bottom chip, and then the liquid-cell was assembled with the windows (50 µm × 200 µm) aligned perpendicularly (50 µm x 50 µm LCTEM viewing area), and the lines of the holder were sealed off without external flow.

Experiments were performed using a JEM-ARM300F (JEOL Ltd., Tokyo, Japan) operated at 300 keV and a JEM-ARM200CF (JEOL, Ltd., Tokyo, Japan) operated at 200 keV. Micrographs were recorded on a 2k × 2k Gatan OneView-IS CCD camera (Gatan Inc., Pleasanton, CA, USA) using Gatan Digital Micrograph image acquisition software (Roper Technologies, Sarasota, FL). The electron flux values used in LCTEM experiments were calculated using the beam current for each aperture selection, as measured by a Faraday Holder through vacuum, and the beam diameter incident upon the sample. Immediately

following LCTEM experiments, the SiN$_x$ chips were carefully separated and allowed to dry.

## Image processing

We performed image processing using the software Fiji (Fig. 3c). First, we cropped a region of interest in a fixed area for each timepoint (Fig. 5a). We binned the cropped images (2 × 2 average), applied a gaussian filter (σ = 1), thresholded each cropped region of interest, and subtracted features less than 5 pixels[2] from each image.

## MALDI-IMS

LCTEM chips, with their SiN$_x$ membranes facing upwards, were adhered to the conductive face of an ITO-coated glass slide with 70–100 ohms resistivity (Bruker Daltonics), using ~0.5 μL nail polish and allowed to dry. To equalize the height difference from SiN$_x$ chips on the slide (~0.25 mm), four pieces of Scotch tape were applied to both short edges of the slide on the same side. As a control, an unimaged polymer solution was dropcasted onto a clean liquid-cell chip. All chips were coated with trans-2-[3-(4-tert-butylphenyl)−2-methyl-2-propenylidene]malononitrile (DCTB) matrix in (10 mg mL$^{-1}$ in acetonitrile).

Slides were mounted into an MTP Slide Adapter II and loaded onto a Bruker Rapiflex MALDI-ToF mass spectrometer for analysis using the flexControl software (Bruker Daltonics). Samples were analyzed by MALDI-MS under reflector positive mode (2000–10000 Da) using a 355 nm smartbeam 3D laser with a 50 μm focus diameter and 200 Hz frequency, a constant laser power of 50%, and a sum of 500 shots per spectrum. Spectra were collected using an accelerating voltage of 20 kV and detector gain of 792 V. Region of interest (ROI) mapping was performed at a raster width of 50 μm, and image analysis was performed in flexImaging software (Bruker Daltonics).

## RSoXS

RSoXS was performed at BL 11.0.1.245 of the Advanced Light Source (ALS) at LBNL using a custom Protochips liquid flow cell. A back-illuminated Princeton PI-MTE CCD cooled to −45 °C detected the scattering pattern with exposures between 30 and 120 s. Next, 0.5 μL of sample was pipetted manually onto the bottom chip, and then the liquid-cell was assembled with the windows (50 μm × 200 μm) aligned perpendicularly (50 μm x 50 μm irradiation area).

Raw RSoXS scattering patterns collected on the 2D area detector were reduced to 1D scattering profiles via azimuthal averaging in the NIKA processing package developed by Jan Ilavsky with a modified handling of the energy dimension via a custom panel58 and processing functions in IGOR Pro 8 (Wavemetrics). This included standard processing for experimental geometry, corrections for energy-dependent incident beam intensity, and dark background subtraction. Data was fit to well-defined form factor models using the SASView 5.0.5 software (http://www.sasview.org/).

## DLS

Dynamic light scattering (DLS) measurements were performed on a Zetasizer (Malvern Instruments Ltd, Nano ZS) with a 120 second equilibrium time for each measurement.

## SEC-MALS

Polymers were dissolved at 5 mg mL-1 in DMF and analyzed by size exclusion chromatography multiangle light scattering (SEC-MALS) with on a Phenomenex Phenogel 5 u 103 Å, 1K-75K, 300 × 7.8 mm in series with a Phenomenex Phenogel 5 u 103 Å, 10K-100K, 300 × 7.80 mm) at 65 °C in 0.05 M LiBr in DMF, using a ChromTech Series 1500 pump equipped with a multi-angle light scattering detector (DAWNHELIOS II, Wyatt Technology) and a refractive index detector (Wyatt Optilab T-rEX) normalized to a 30000 MW polystyrene standard at a flow rate of 0.75 mL/min.

## ¹ H NMR

Proton Nuclear Magnetic Resonance Spectroscopy (1 H NMR) experiments were all performed on a Bruker Advance III HD system equipped with a TXO Prodigy probe.

## Data availability

We declare that all other data supporting the findings of this study are available within the article and Supplementary Information files. The source data are available in the figshare repository https://doi.org/10.6084/m9.figshare.22775696 and are also available from the corresponding author upon request.

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

## Acknowledgements

Research in the N.C.G. group was conducted with government support under and awarded by DoD through the ARO (W911NF-17-1-0326). In addition, the N.C.G. thanks the NSF for support through a joint research grant (CHE-MSN 1905270). This work was also supported by a MURI grant (W911NF-15-1-0568). This research used EPIC facility of Northwestern University's NUANCE Center, which has received support from the Soft and Hybrid Nanotechnology Experimental (SHyNE) Resource (NSF ECCS-1542205), the MRSEC program (NSF DMR1720139) at the Materials Research Center; the International Institute for Nanotechnology (IIN), the Keck Foundation, and the State of Illinois, through the International Institute for Nanotechnology. This research used resources of the Advanced Light Source, a U.S. DOE Office of Science User Facility under contract no. DE-AC02-05CH11231. J.K. gratefully acknowledges support from the Ryan Fellowship and the International Institute for Nanotechnology at Northwestern University.

## Author contributions

J.K. and N.C.G. devised the project. J.K. performed polymer synthesis, LCTEM, MALDI-IMS, created a kinetic model for IPA radiolysis, and drafted the paper. C.W. performed liquid RSoXS. All authors wrote and edited the manuscript. All authors have given approval to the final version of the manuscript.

## Competing interests

The authors declare no competing interests.
