## [Peer Review File · Nature Communications]

Upper Critical Solution Temperature Polymer Assemblies via Variable Temperature Liquid Phase Transmission Electron Microscopy and Liquid Resonant Soft X-Ray ScatteringReviewers' Comments:

Reviewer #1:

Remarks to the Author:

In their manuscript, Korpanty et al investigate the nanophase thermal behavior of triblock copolymer micelles in-situ to isopropanol (IPA). The mid-block is a thermoresponsive polymer that contains an upper critical solution temperature (UCST) interaction with the solvent. They utilize temperature dependent light scattering, electron microscopy (TEM), and resonant scattering (RSoXS) and report core mixing of the mid-block below UCST and irreversible expulsion of the mid-block to the corona above UCST with a reversible shrinking of the overall micelle size with increased temperature.

The use of similar microfluidic cells for the multimodal measurements is relatively novel and enables characterization of the behavior under similar confinement conditions. Such a measurement combination could be ground-breaking in investigating the dynamics of self-assembled polymer nanostructures (of interest in a variety of applications) because of the extra chemical sensitivity provided by resonant scattering. The synthesis route using polymerization induced self-assembly for a UCST-type polymer in IPA appears to be novel as well, although I'm not a polymer chemist.

The issue keeping me from recommending this manuscript for publication is the general lack of rigor in supporting the claims with evidence. In particular the UCST thermoresponsive mixing behavior is not supported at all, and even the reversible size change is only really demonstrated well in the rather mundane light scattering measurements.

The issues start in the e-beam damage study. First it is not clear what "mass signal intensity" (pg 6, used to quantify damage) is in the first place. Is this just transmitted electron intensity? Second, it is not clear what I_c is ("intensity of an unimaged control"). I presume it takes a dose to image I_c , so how can this be 'unimaged'? Much more experimental procedures/details are necessary. Finally, and most importantly, it appears the lower of the two doses (0.33 e-/Å/s) still highly damages the polymer where the I_s/I_c ratio is 0.5 for the mid-block and solvophilic block. Thus I contend that the authors have not established "safe imaging conditions" and that their images are likely still severely damaging the sample. It's perhaps not creating a film as in the higher flux, but there could be other artifacts. In particular a major claim is that the size decreases with temperature (Fig 3) and is reversible. A rigorous control series where the size doesn't change over with a similar dose is required to believe this isn't a damage effect. Also, a series where the temperature is again reduced and the size returns is required.

Next the core demixing reported using the TEM "false color filter" is not believable. How does one take a gray scale pixel and assign it three RGB values? Even given such an algorithm, how could this possibly be converted to different polymer chemical components and separated from thickness variations?

For the RSoXS experiment, it is disappointing that no analogous damage check is done. This is especially important since there is no flow and beam damage will be cumulative over the course of the experiment. Indeed, much more detail on the actual experiment procedure/method is required. In the RSoXS analysis, much more clarity is required regarding the model and fits. Is the "broad peak form factor" simply representing the size polydispersity; what kind? The precise fit function needs to be explicitly described or referenced somewhere as well as all parameters, their final values, and how the fits are conducted.

How is the "microphase separation" parameter in the model? This is important because it is claimed that this is sensitive to chemical mixing inside the core. Another problem is the claimed size of this separation, which is on the order of Angstroms to a few nm. This is not believable given the q -range measured in the data ending at $2\pi/q_{max} \sim 30\text{nm}$. In fact, at the high- q range, the model dives down in intensity. This suggests that the fit parameters resulting in the smallest features are not real (fitting

noise features) and calls into question all of the reported parameter values.

The energy-dependent scattering profiles are impressive, however chemical information cannot be extracted without using optical models. Vaguely referring to different energies as absorbing by certain moieties is an unacceptable oversimplification, since scattering doesn't simply come from absorption but from dispersion as well.

Based on these deficiencies, there is unfortunately no evidence to support the dynamic model proposed in figure 5. Without this, there is still enough novelty for Nature Communications, but other critical aspects described above need to be resolved/reinforced. In particular, all three primary measurements (light scattering, TEM, RSoXS) should be able to consistently show a reversible size change with temperature – something I don't see in TEM or RSoXS.

Smaller things:

- I can't make heads or tails of Figure S4 or S5 (MALDI-IMS). Was there supposed to be a discussion somewhere?
- Why were only two fluxes used in the damage assessment of TEM? How were they chosen?
- Why are there 91 more references in the SI? Where do they come from?
- The videos appear to all be damage experiments at the high flux. Why no control experiments at the low flux? Also in several videos, nothing appears to be happening, so I don't know what the point is.

Reviewer #2:

Remarks to the Author:

The manuscript describes UCST behavior in solutions of thermally responsive triblock copolymers. The structure and thermally responsive behavior of triblock surfactants are complex, their characterization is challenging, and new measurement approaches are valuable to the community. The characterization suite applied here is elegant and cutting-edge, consisting of liquid-cell TEM and RSoXS, both with in-situ temperature control. Although the general theme of the content is certainly suitable for this fine journal, and there are some important "firsts" here, there are many opportunities for refinement of the message and for enhancing the technical approaches. I would urge the editor to permit revision for further consideration.

1) Although the proposed real-space / reciprocal space framework is certainly elegant, the manuscript lacks a clear articulation of what new capability these techniques or this combination of techniques bring to the table. Lots of language is used in the manuscript to describe the power of approach: "Unique insight" [abstract], "powerful," [p3], "real space and reciprocal space insight," [p3], "obtained key insights into the complex structure and dynamics," [p13]. But it is never clearly articulated how this combination of approaches provides information that could not be determined with more conventional solution SAXS/SANS approaches.

1a) With regard to the combination of real & reciprocal, for example, the authors could discuss:

- Uniqueness issues commonly encountered by applying scattering alone, how having real space images makes for more confident model selection & fitting constraints.

- Issues of whether the small measurement volume of microscopy alone would be representative.

Having RSoXS of a larger volume enhances confidence that the characterization is representative of bulk solution.

1b) With regard to RSoXS over conventional SANS, there is of course its advantages in not requiring radiolabeling and its preference for small volumes. These advantages were already heavily discussed and demonstrated in this fine journal in McAfee, et al. Nature Communications 12, 3123 (2021). However, I believe this might be the first example of solution RSoXS in an organic solvent, and its application here is potentially an important demonstration that such an experiment is possible at all considering that the solvent would have a significant cross-section at energies $> \sim 287$ eV. This latter point may be worth mentioning!

1c) With regard to in-situ TEM, is there anything special beyond beam damage that is learned from the video frame rate capabilities? They are certainly very exciting. "Dynamics" are mentioned a couple of times but there is no quantitative analysis of dynamics. I suppose that the evolution of the structure in both time and temperature is revealed here and there is no necessity to assume quasi-steady state behaviors at different temperatures. Perhaps some of these advantages should be amplified.

1d) Ultimately the key new observation that the authors report: "a UCST-triggered change in morphology from a polydisperse core-shell structure with a complex core to a more-ordered spherical micelle," [p 13] would seem to this reviewer to be straightforwardly discoverable with conventional approaches so again I urge the authors to provide a more complete explanation on why its discovery is owed to such a novel characterization approach.

2) RSoXS is a powerful emerging characterization method, and as a community we are just beginning to scratch the surface of its transformative capabilities. I provide comments here to encourage its rigorous discussion and application in a manner that would better acknowledge conventions and best-practices from solution SAXS and SANS. To this end, I have the following suggestions:

2a) Most SAXS and SANS practitioners would not attempt to fit data that were not rigorously background subtracted. Can that be done with this experimental setup?

2b) The RSoXS is fit to pure form factor expressions when the TEM indicates a significant structure factor should be present, with agglomerates of spherical particles.

2c) Is there enough information in the RSoXS pattern to be confident in fitting multiple parameters to the single feature? It could be a Guinier-Porod knee, a structure factor peak, or a form factor maximum. If this were solution SANS, I would insist that the data are significantly overfit.

2d) Framed differently, there is no uncertainty analysis that supports the fitting, and I am uncomfortable that several features are beyond the diffraction limit of the radiation used. My confidence would be increased were an uncertainty analysis provided showing that the multiparameter fit has low uncertainty.

3) A separate issue that should be addressed is that beam damage is assessed extensively in the TEM experiment (see all the videos), but doesn't appear to be assessed in the RSoXS experiment. In McAfee, et al. *Nature Communications* 12, 3123 (2021), the authors there assert that a flow cell is used specifically to prevent beam damage effects, and such effects may be significant as soft X-rays are ionizing radiation. The cells used here are not flow cells, they are quiescent. Could the authors please provide evidence that X-ray beam damage does not occur in their system? Perhaps sequential short exposures showing that the pattern is roughly the same for early and late exposures on the same spot would be sufficient.

4) If we were to characterize the core-shell assembly that this triblock makes in solution using conventional SANS approaches, we would likely use contrast-variation SANS with monotonic deuteration of one or more blocks and the extraction of partial scattering functions relevant to single blocks that could be fit separately. This is of course another key advantage of RSoXS in that the SANS experiment would require REALLY extensive deuteration because you might need two different monotonic deuteration schemes to isolate three partial scattering functions. With all of that said, I find the discussion of energy selection in RSoXS and its ability to highlight different materials to be insufficiently quantitative and appearing to suggest incorrect underlying principles of the technique. I am sure the authors didn't intend this; Cheng Wang is one of the pioneers of the technique! However, it should be addressed so that the new technique can put its best foot forward.

Three energies were used: 283 eV, 288.8 eV, and 285.2 eV. 283 eV is described simply as "[the energy that] showed the most significant scattering intensity," 285.2 is said to "probe the PS core," and 288.8 is said to "evaluate the PEG corona." The latter two assertions cite earlier work, and lean on a premise advanced in those earlier works that the selection of energies near key NEXAFS resonances is sufficient to measure a scattering function that is interpretable in ways similar to a partial scattering function from contrast-variation SANS approaches. That is to say, there is an implicit assumption made by the authors that by moving to an energy where one phase in a multiphase sample has

significant absorbance, the less-absorbing phases are 1) contrast-matched to each other, and 2) having the same contrast with the highlighted phase. Although this premise was certainly sufficient to convey the promise of the technique in its early days and motivate further investment in understanding it, it ignores two key factors A) that contrast is developed by both beta (absorbance) and $\Delta\epsilon$, and B) contrast is a matter of differences, not absolute intensities (or even necessarily peaks) in beta or $\Delta\epsilon$.

Recent work by the Brian Collins group (Ferron et al., Physical Review Letters 119, no. 16 2017: 167801., Figure 1 caption) has demonstrated that a more quantitative framework is required to discuss these physical principles and how they relate to the validity of such assumptions. Ferron et al. describe binary contrast calculations incorporating differences in both beta and $\Delta\epsilon$, and this framework is more consistent with best-practices in SAXS and SANS, in which contrast is always understood to be based on differences (and in those techniques is due mostly to $\Delta\epsilon$, not beta!). One way to apply that modern understanding of RSoXS principles to the work under review would be to first calculate binary contrast functions among the 3 blocks from reference dielectric functions in both beta and $\Delta\epsilon$. There will be $4!/2 = 12$ binary contrast functions, but it depends on whether you consider IPA implicit and also whether you include various degrees of swelling; I would leave that up to the authors. Considering only the 3 "solid"-solid contrasts, for 285.2 eV it will be important to show that the PEG vs. PS and PEGMeA-co-PEGPhA vs. PS binary contrasts are of similar magnitude and both much, much larger than the PEG vs. PEGMeA-co-PEGPhA contrast (because there is an implicit assumption that these are contrast-matched at that energy when solving for the PS core). For 288.8 eV, similarly, it will be important to show that the PEG vs. PEGMeA-co-PEGPhA and PEG vs PS contrasts are similar, and very large relative to the PEGMeA-co-PEGPhA vs. PS contrast. What I have proposed does not require additional experiments (I would cobble together the PEGMeA-co-PEGPhA dielectric function from weighted reference functions in Ade's library <https://www.physics.ncsu.edu/stxm/polymerspectro/> or Gann's <https://github.com/EliotGann/Optical-Constants-Database>), and it is perhaps an hour of additional computation to prove that these assumptions are valid.

Modifying the RSoXS discussion and making its assumptions regarding energy selection more quantitative would put the RSoXS discussion on firmer footing, dispel questions about how the technique might compare to contrast-variation SANS, and potentially place an even stronger spotlight on the exciting new capabilities this technique brings to soft matter structural characterization.

Reviewer #1 (Remarks to the Author):

In their manuscript, Korpanty et al investigate the nanophase thermal behavior of triblock copolymer micelles in-situ to isopropanol (IPA). The mid-block is a thermoresponsive polymer that contains an upper critical solution temperature (UCST) interaction with the solvent. They utilize temperature dependent light scattering, electron microscopy (TEM), and resonant scattering (RSoXS) and report core mixing of the mid-block below UCST and irreversible expulsion of the mid-block to the corona above UCST with a reversible shrinking of the overall micelle size with increased temperature.

The use of similar microfluidic cells for the multimodal measurements is relatively novel and enables characterization of the behavior under similar confinement conditions. Such a measurement combination could be ground-breaking in investigating the dynamics of self-assembled polymer nanostructures (of interest in a variety of applications) because of the extra chemical sensitivity provided by resonant scattering. The synthesis route using polymerization induced self-assembly for a UCST-type polymer in IPA appears to be novel as well, although I'm not a polymer chemist.

The issue keeping me from recommending this manuscript for publication is the general lack of rigor in supporting the claims with evidence. In particular the UCST thermoresponsive mixing behavior is not supported at all, and even the reversible size change is only really demonstrated well in the rather mundane light scattering measurements.

The issues start in the e-beam damage study. First it is not clear what "mass signal intensity" (pg 6, used to quantify damage) is in the first place. Is this just transmitted electron intensity?

We thank the reviewer for their comments. The mass signal is obtained from the MALDI-IMS mass spectral experiment. We refer here to the intensity of the signal of a given m/z peak. This is unrelated to electron intensity and is performed after the conclusion of the LCTEM experiment as an entirely separate measurement. To ensure this is made clear, we have added the following to the text:

"Upon completion of LCTEM damage experiments, we studied the extent of polymer degradation via a separate matrix-assisted laser desorption/ionization imaging mass spectrometry (MALDI-IMS) experiment.^{32, 35} Here, MALDI-IMS was used to evaluate the mass signal intensity of the polymer distribution over the imaged area. Because the triblock copolymer does not efficiently ionize,³² we performed these LCTEM and MALDI-

IMS damage experiments on lower molecular weight proxies for the UCST triblock copolymer (**Figures S4-S6**)."

Second, it is not clear what I_c is ("intensity of an unimaged control"). I presume it takes a dose to image I_c , so how can this be 'unimaged'? Much more experimental procedures/details are necessary.

We thank the reviewer for pointing out the need to clarify terminology. I_c denotes the mass signal intensity for an unimaged control. In line with our previous studies, the unimaged control was prepared by dropcasting an aliquot of the polymer onto a liquid-cell chip and then performing the MALDI-IMS experiment to generate a mass spectral map of the liquid-cell chip surface. Here, the sample was not imaged by LCTEM but rather directly subjected to MALDI-IMS without imaging by electron microscopy. Thus, no electron dose was used to assess the sample because no LCTEM experiment was performed. To ensure this is made clear, we have added the following:

"Data from these experiments were used to develop beam damage diagrams where the ratio of the mass signal intensity for different imaged samples (I_s) to the intensity of an unimaged control (I_c) was plotted as a function of the instantaneous flux and cumulative fluence (**Figure 2b**). The unimaged control sample was prepared by dropcasting the same polymer solution used in LCTEM damage experiments onto a liquid-cell chip and then immediately conducting MALDI-IMS."

Finally, and most importantly, it appears the lower of the two doses (0.33 e-/A/s) still highly damages the polymer where the I_s/I_c ratio is 0.5 for the mid-block and solvophilic block. Thus I contend that the authors have not established "safe imaging conditions" and that their images are likely still severely damaging the sample. It's perhaps not creating a film as in the higher flux, but there could be other artifacts. In particular a major claim is that the size decreases with temperature (Fig 3) and is reversible. A rigorous control series where the size doesn't change over with a similar dose is required to believe this isn't a damage effect. Also, a series where the temperature is again reduced and the size returns is required.

We thank the reviewer for their comment. It is impossible to avoid sample damage as irradiating the sample with the electron beam inherently perturbs the chemistry of the material under investigation; as with any microscopy experiment. Our goal was to limit and quantify damage, which we believe we accomplished since the majority ($I_s/I_c > 0.5$) of each proxy polymer shown in the damage plot in Fig 2b survives. Moreover, we believe

that our results reveal how the different polymers undergo damage. Reproduced from the SI in the text below, we analyze the radiolytic sensitivity of different polymers.

“As revealed by MALDI-IMS, PEG-*b*-PEGMeA-*co*-PEGPhA and PEGMeA remained relatively intact under low flux conditions and showed reduced I_s/I_c values under high flux conditions, indicating enhanced sample degradation under higher flux conditions (**Figures 2b, S4-S6**). Interestingly, the PDEGEA polymer showed similar signal intensity under low and high flux conditions, suggesting that damage to the OEG sidechains is the dominant form of electron beam damage for these acrylic polymers, rather than damage to the polymer backbone. Since degradation of the OEG sidechains should reduce the solvophilicity of the polymer, damage to these sidechains could be a potential cause of the observed film formation under continuous LCTEM imaging (**Figure 2a**). Likewise, as PS showed enhanced degradation under continuous imaging, crosslinking of PS polymer chains could also increase solvophobicity and thus contribute to the observed film formation. Ultimately, our MALDI-IMS studies suggest that the triblock copolymer likewise survives under low flux conditions, whereas under high flux conditions, the particle expansion is a manifestation of sample damage (**Figures 2a, 3a**).”

Additionally, we agree that we need to perform this crucial control experiment to show that particle size reduction happens only in the presence of heating. We have done so, and this experiment is shown below and has been added to the SI, along with the following sentence that has been added to the main text:

“Moreover, under low flux stroboscopic conditions with no heating, the nanoassemblies did not change in size, further suggesting that the UCST transition is indeed responsible for the observed particle contraction (**Figure S10**).”

Figure S10. Control LCTEM experiment imaged at $0.33 \text{ e}^- \text{ \AA}^{-2} \text{ s}^{-1}$. Control experiment of unheated UCST polymer showing electron beam dose not trigger particle size reduction.

We also point the reviewer to control experiments already in the SI, where we show that even in the absence of the electron beam, with only an initial timepoint taken before heating and then another timepoint taken again only upon cooling, the UCST transition is irreversible. This is a feature that we also observed in RSoXS experiments (Figure 4b) and in DLS data (Figure 1d). The irreversibility of this transformation is likely due to the nanoassemblies being kinetically trapped. In a similar system with an LCST transition, we have previously observed the same irreversibility (Gianneschi et al., *Nature Communications* **2021**, 12 (1), 6568). Moreover, in general, as polymers tend to strongly adhere to the silicon nitride membrane, transformations observed by LCTEM tend to be irreversible in many of our published studies.

Figure S10. Control LCTEM experiments all imaged at $0.33 \text{ e}^- \text{ \AA}^{-2} \text{ s}^{-1}$. a. Heating and cooling cycle VT-LCTEM experiment showing the irreversibility of the UCST transformation. b. Post-mortem dry state of LC chip showing uniform small assemblies c. Final, cooled timepoint of data shown in Figure S3 showing irreversibility of the UCST transformation.

Next the core demixing reported using the TEM “false color filter” is not believable. How does one take a gray scale pixel and assign it three RGB values? Even given such an algorithm, how could this possibly be converted to different polymer chemical components and separated from thickness variations?

The use of such false color filters is relatively common e.g. Patterson et al., *ACS Nano* **2021**, *15* (6), 10296-10308. Indeed, we have used it in prior publications: Gianneschi et al., *Nature Communications* **2021**, *12* (1), 6568 and Gianneschi et al., *Cell Reports Physical Science* **2022**, 100772. As in previous examples, this false color processing was achieved using the “lookup table” feature in ImageJ/Fiji. We specifically used the 6-shade filter. To make this clear, we have added the statement below to the main text. While liquid thickness variations may affect the false color filter results, this should only affect the background, not the contrast within the individual nanostructures (as discussed in our prior work extensively in Gianneschi et al., *Nature Communications* **2021**, *12* (1), 6568). We agree with the reviewer that polymer chemical components cannot be measured/determined using the false color filter. Rather, the false color filter is a sensitive method for looking at the density of different components within the sample, which we use to infer the solvophilicity of different colored regions, not the type of polymer block in a given location. In this work we couple real space LCTEM imaging with RSoXS to shed light on the nature of the blocks, and to track electron density changes.

“To gain insight into the UCST transition, we applied a background subtraction, 4 x 4 average binning, and a false color filter (“6-shade” lookup table) by which each gray scale pixel was assigned a corresponding RGB value.”

For the RSoXS experiment, it is disappointing that no analogous damage check is done. This is especially important since there is no flow and beam damage will be cumulative over the course of the experiment. Indeed, much more detail on the actual experiment procedure/method is required.

While a previous study by McAfee et al. showed that a damage analysis for liquid RSoXS could be done, McAfee used a flow cell for RSoXS experiments to aid in such a beam damage check (McAfee et al. Label-free characterization of organic nanocarriers reveals persistent single molecule cores for hydrocarbon sequestration. *Nat Commun* **12**, 3123 (2021). <https://doi.org/10.1038/s41467-021-23382-8>). In contrast to McAfee’s study, our materials are preformed and thus cannot be flowed in using current liquid RSoXS platforms, preventing such a damage analysis from being performed. Moreover, beam damage in TEM experiments was largely assessed by coupling *post-mortem* MALDI-IMS with LCTEM. These experiments were specifically performed for lower molecular weight proxies for each block within the UCST polymer. Finally, performing *post-mortem* MALDI-IMS following liquid RSoXS would require a MALDI-IMS instrument to be available onsite at LBNL, which currently is not the case.

In the RSoXS analysis, much more clarity is required regarding the model and fits. Is the “broad peak form factor” simply representing the size polydispersity; what kind? The precise fit function needs to be explicitly described or referenced somewhere as well as all parameters, their final values, and how the fits are conducted.

We agree that we should improve our discussion of the RSoXS data. We have added more detail to the SI and have clearly stated how the fits were conducted and what values were used to produce the fits. We also agree that we need to be clearer on how our model was constructed and upon reading all of the reviewer’s comments, we have decided to alter our model to avoid summing the broad peak form factor with a core-shell model, a model that we were initially motivated to use because of our previous studies on a similarly complex, thermoresponsive multi-block copolymer (Gianneschi et al., *Nature Communications* **2021**, 12 (1), 6568). The reviewer’s comments motivated us to explore a simpler model, and we ultimately found that a polymer micelle model could better describe our system and we applied this model to all measurements to enable easier comparison between measurements. Below, we have reproduced the discussion in the main text.

“Prior to conducting RSoXS measurements, we performed NEXAFS spectroscopy measurements on pure PS, PEGMeA-co-PEGPhA, and PEG films to generate binary contrast functions for the polymers (**Supplementary Figure 13**). At an energy of 283 eV, the contrast between PEGMeA-co-PEGPhA and PEG is small, while the contrast between PS and these two polymers is large. Thus, we used an energy of 283 eV to probe the overall core-shell(-shell) morphology of the PEG-*b*-PEGMeA-co-PEGPhA-*b*-PS polymer structures, as this energy showed the most significant scattering intensity for energies near the carbon K-edge (**Figures 4a, Supplementary Figure 14**).²⁷ Using an energy of 283 eV, at room temperature, a broad trace was observed, indicative of the high polydispersity of the initial assemblies before heating. The room temperature data was fit to a spherical polymer micelle form factor, a morphological model informed by liquid-phase real space imaging. Using this fit revealed that, on average, the initial assemblies had a core radius of 47 nm with a coronal thickness of 48 nm. Upon heating the sample to 60 °C for 10 minutes, an increase in scattering intensity was observed, indicating ordering of the nanoassemblies, in agreement with VT-DLS data showing a tightening of the distribution. Fitting this trace to a polymer micelle form factor showed a core radius of 42 nm and coronal radius of 59 nm. The decrease in the size of the core can be explained by the enhanced solubility of the PEGMeA-co-PEGPhA block upon the UCST transition. Presumably, at room temperature, the core is PS and PEGMeA-co-PEGPhA-rich, as PEG is the only solvophilic block at room temperature and should thus be found in the outer shell. However, upon heating, PEGMeA-co-PEGPhA becomes increasingly solvophilic,

leading to an effective increase of the solvophilic corona and a decrease in the insoluble core. This outer corona is “invisible” by LCTEM since this shell is entirely soluble in isopropanol (**Figure 3c**). After being held for 5 additional minutes at 60 °C, we observed a further increase in the scattering intensity. We applied a polymer micelle form factor, yielding a fit with a core radius of 39 nm with coronal thickness of 73 nm, demonstrating further expulsion of PEGMeA-co-PEGPhA from the particle core, consistent with TEM imaging data (**Supplementary Figure 10**). By RSoXS, the nanoassemblies remained in this morphology after being held at 60 °C for 15 more minutes, with a coronal radius of 75 nm and core radius of 38 nm. Upon cooling and holding the sample at < 35 °C for 30 minutes, the sample was fit to a polymer micelle form factor with a core radius of 38 nm with coronal thickness of 65 nm, demonstrating partial reversibility of the UCST transformation given the decrease in coronal thickness.

To further probe the final, trapped morphology after the heating-cooling cycle, we conducted energy dependent RSoXS at 283 eV, 285.2 eV, and 288.8 eV (**Figure 4b**). While the 283 eV data describes the overall micellar morphology, using an energy of 285.2 eV allowed us to enhance contrast from the PS domains. Using a spherical form factor with a sticky hard sphere structure factor, the fit for the 285.2 eV trace revealed a 21 nm PS core radius. As the 283 eV fit shows that the entire core radius was 48 nm, the smaller radius measured at 285.2 eV suggests that, upon cooling, the core likely contains both PS and PEGMeA-co-PEGPhA, pointing to a partial reversibility of the UCST transition.

Using an energy of 288.8 eV results in similar contrast for all three polymer domains, allowing us to probe the overall particle size. Given the similarity in contrast for all three domains, the 288.8 eV trace showed low scattering contrast, and so the overall scattering can be more easily observed by plotting $\ln(I \times Q^2)$ vs $\ln Q$ plot (**Supplementary Figure 15**). Additionally, the low scattering contrast may partially be attributed to our inability to perform a solvent background subtraction due to the variability of liquid thickness over separate experiments. In future studies, this issue may be ameliorated by creating a flow cell capable of flowing preformed nanomaterials for liquid RSoXS measurements. Nonetheless, fitting the 288.8 eV trace to a spherical form factor with a sticky hard sphere structure factor showed an overall sphere diameter of 130 nm. It is worth noting that for the energies that we have investigated near the carbon K-edge, the isopropanol solvent itself has a strong cross-section. This consideration has likely prevented others from pursuing liquid RSoXS studies in organic solvents, making our study the first of its kind. However, even without background subtraction, we were still able to obtain sufficient signal intensity to probe the UCST transition, demonstrating the utility of liquid RSoXS to investigate non-aqueous systems in future studies.”

How is the “microphase separation” parameter in the model? This is important because it is claimed that this is sensitive to chemical mixing inside the core. Another problem is the claimed size of this separation, which is on the order of Angstroms to a few nm. This is not believable given the q -range measured in the data ending at $2\pi/q_{\max} \sim 30\text{nm}$. In fact, at the high- q range, the model dives down in intensity. This suggests that the fit parameters resulting in the smallest features are not real (fitting noise features) and calls into question all of the reported parameter values.

We thank the reviewer for their feedback. Please see comments above. We have corrected this and hope the reviewer finds the new discussion acceptable.

The energy-dependent scattering profiles are impressive, however chemical information cannot be extracted without using optical models. Vaguely referring to different energies as absorbing by certain moieties is an unacceptable oversimplification, since scattering doesn't simply come from absorption but from dispersion as well.

We thank the reviewer for their comment. We have provided more detail on the chemical sensitivity of RSoXS measurements, as shown below.

“Prior to conducting RSoXS measurements, we performed NEXAFS spectroscopy measurements on pure PS, PEGMeA-*co*-PEGPhA, and PEG films to generate binary contrast functions for the polymers (**Supplementary Figure 13**). At an energy of 283 eV, the contrast between PEGMeA-*co*-PEGPhA and PEG is small, while the contrast between PS and these two polymers is large. Thus, we used an energy of 283 eV to probe the overall core-shell(-shell) morphology of the PEG-*b*-PEGMeA-*co*-PEGPhA-*b*-PS polymer structures, as this energy showed the most significant scattering intensity for energies near the carbon K-edge”

“To further probe the final, trapped morphology after the heating-cooling cycle, we conducted energy dependent RSoXS at 283 eV, 285.2 eV, and 288.8 eV (**Figure 4b**). While the 283 eV data describes the overall micellar morphology, using an energy of 285.2 eV allowed us to enhance contrast from the PS domains. Using a spherical form factor with a sticky hard sphere structure factor, the fit for the 285.2 eV trace revealed a 21 nm PS core radius. As the 283 eV fit shows that the entire core radius was 48 nm, the smaller radius measured at 285.2 eV suggests that, upon cooling, the core likely contains both PS and PEGMeA-*co*-PEGPhA, pointing to a partial reversibility of the UCST transition.

Using an energy of 288.8 eV results in similar contrast for all three polymer domains, allowing us to probe the overall particle size. Given the similarity in contrast for all three domains, the 288.8 eV trace showed low scattering contrast, and so the overall scattering can be more easily observed by plotting $\ln(I \times Q^2)$ vs $\ln Q$ plot (**Supplementary Figure 15**). Additionally, the low scattering contrast may partially be attributed to our inability to perform a solvent background subtraction due to the variability of liquid thickness over separate experiments. In future studies, this issue may be ameliorated by creating a flow cell capable of flowing preformed nanomaterials for liquid RSoXS measurements. Nonetheless, fitting the 288.8 eV trace to a spherical form factor with a sticky hard sphere structure factor showed an overall sphere diameter of 130 nm. It is worth noting that for the energies that we have investigated near the carbon K-edge, the isopropanol solvent itself has a strong cross-section. This consideration has likely prevented others from pursuing liquid RSoXS studies in organic solvents, making our study the first of its kind. However, even without background subtraction, we were still able to obtain sufficient signal intensity to probe the UCST transition, demonstrating the utility of liquid RSoXS to investigate non-aqueous systems in future studies.”

Supplementary Figure 13. NEXAFS data and binary contrast function. a. Binary contrast functions between different homopolymers. **b.** NEXAFS measurements for homopolymers and UCST triblock copolymer.

Based on these deficiencies, there is unfortunately no evidence to support the dynamic model proposed in figure 5. Without this, there is still enough novelty for Nature Communications, but other critical aspects described above need to be resolved/reinforced. In particular, all three primary measurements (light scattering, TEM, RSoXS) should be able to consistently show a reversible size change with temperature – something I don't see in TEM or RSoXS.

We thank the reviewer for all of their thorough comments and hope the reviewer finds our changes to be satisfactory.

Smaller things:

- I can't make heads or tails of Figure S4 or S5 (MALDI-IMS). Was there supposed to be a discussion somewhere?

We thank the reviewer for their comment. Figure S4 is the mass-filtered colormap of the liquid-cell chips and S5 shows the corresponding MALDI mass spectra. These data were used to create the damage plot. The discussion associated with these figures is shown below, and we have added more information to clarify what these figures denote:

“For LCTEM damage experiments performed on proxy polymers, we used MALDI-IMS to create mass-filtered maps of the liquid-cell chip surfaces (**Figure S4**). From this data, we obtained average mass spectra of the liquid-cell chip surfaces, which we then used to assess the MALDI mass signal intensity of the sample under different imaging conditions (**Figure S5**). As revealed by MALDI-IMS, PEG-*b*-PEGMeA-*co*-PEGPhA and PEGMeA remained fairly intact under low flux conditions and showed reduced I_s/I_c values under high flux conditions, indicating enhanced sample degradation under higher flux conditions (**Figures 2b, S4-S6**). Interestingly, the PDEGEA polymer showed similar signal intensity under low and high flux conditions, suggesting that damage to the OEG sidechains is the dominant form of electron beam damage for these acrylic polymers, rather than damage to the polymer backbone. Since degradation of the OEG sidechains should reduce the solvophilicity of the polymer, damage to these sidechains could be a potential cause of the observed film formation under continuous LCTEM imaging (**Figure 2a**). Likewise, as PS showed enhanced degradation under continuous imaging, crosslinking of PS polymer chains could also increase solvophobicity and thus contribute to the observed film formation. Ultimately, our MALDI-IMS studies suggest that the triblock copolymer likewise survives under low flux conditions, whereas under high flux conditions, the particle expansion is a manifestation of sample damage (**Figures 2a, 3a**).”

- Why were only two fluxes used in the damage assessment of TEM? How were they chosen?

We chose $0.33 \text{ e}^- \text{ \AA}^{-2} \text{ s}^{-1}$ because at this low flux, we were able to clearly see the nanostructures without any clear real space damage effects. In a separate experiment, upon increasing the flux to $1.42 \text{ e}^- \text{ \AA}^{-2} \text{ s}^{-1}$, we instantly observed particle expansion, a clear manifestation of real space damage and so we chose this as our high flux condition.

In previous studies, we also examined two fluxes to converge on optimized imaging conditions and to compare the stability of a given polymer under low flux stroboscopic imaging versus high flux continuous imaging (Gianneschi et al., *Nature Communications* **2021**, 12 (1), 6568, Gianneschi et al., *Cell Reports Physical Science* **2022**, 100772, and Gianneschi, N. C., *Nano Letters* **2021**, 21 (2), 1141-1149).

- Why are there 91 more references in the SI? Where do they come from?

These references were used to create models for isopropanol and water radiolysis. These references list G-values, rate constants, and discuss the radiolytic damage products produced from each solvent. In the SI, we insert these citations in supplementary tables.

- The videos appear to all be damage experiments at the high flux. Why no control experiments at the low flux? Also in several videos, nothing appears to be happening, so I don't know what the point is.

We did not perform low flux continuous imaging damage experiments and rather performed low flux damage experiments under stroboscopic imaging (Figure S6). The reason for this was these same low flux/fluence conditions were applied image the UCST nanoassemblies. Here, continuous imaging would have likely led to polymer damage even under low flux, given the slow kinetics (~2 hours) of the UCST transition (Figure 3). In brief, we did perform low flux damage experiments, but not under continuous imaging conditions. In the higher flux damage experiment videos, often nothing happens because, with the exception of the damage experiment performed on the UCST nanoassemblies, we were examining soluble polymers. Here, we were not motivated to observe real space damage effects and rather, we aimed to assess the integrity of the different polymers after imaging via MALDI-IMS. These data were then used to create the damage plot to compare the stability of the different polymers under low flux stroboscopic imaging and high flux continuous imaging.

Reviewer #2 (Remarks to the Author):

The manuscript describes UCST behavior in solutions of thermally responsive triblock copolymers. The structure and thermally responsive behavior of triblock surfactants are complex, their characterization is challenging, and new measurement approaches are valuable to the community. The characterization suite applied here is elegant and cutting-edge, consisting of liquid-cell TEM and RSoXS, both with in-situ temperature control. Although the general theme of the content is certainly suitable for this fine journal, and

there are some important “firsts” here, there are many opportunities for refinement of the message and for enhancing the technical approaches. I would urge the editor to permit revision for further consideration.

1) Although the proposed real-space / reciprocal space framework is certainly elegant, the manuscript lacks a clear articulation of what new capability these techniques or this combination of techniques bring to the table. Lots of language is used in the manuscript to describe the power of approach: “Unique insight” [abstract], “powerful,”[p3], “real space and reciprocal space insight,”[p3], “obtained key insights into the complex structure and dynamics,” [p13]. But it is never clearly articulated how this combination of approaches provides information that could not be determined with more conventional solution SAXS/SANS approaches.

We entirely agree with the reviewer that we should more rigorously describe the drawbacks of alternative scattering techniques more thoroughly. Accordingly, we have added the following to the main text:

“In recent work, to complement insight into transformations directly observed by VT-LCTEM, VT-SAXS was utilized. While VT-SAXS is a powerful, bulk technique for characterizing thermoresponsive nanomaterials, SAXS scattering contrast relies on differences in electron density between the sample and solvent at high photon energies (~2500 eV). The use of hard X-rays in SAXS provides limited chemical insight into scattering objects and requires that samples be measured in capillaries, potentially posing difficulties in correlating real space LCTEM data with X-ray scattering data due to their distinct sample geometries. Unlike SAXS, small angle neutron scattering (SANS) can enable contrast-matching to selectively analyze scattering from certain parts of a given nanostructure. However, to achieve contrast-matching, the sample must be modified via radiolabeling, requiring additional synthesis that may not be straightforward for every system. Moreover, the low flux of neutron sources compared to X-ray sources requires longer measurement times to obtain sufficient statistics. Like SAXS, SANS also differs greatly in sample geometry from LCTEM, potentially posing difficulties in correlating the two measurements.”

1a) With regard to the combination of real & reciprocal, for example, the authors could discuss:

- Uniqueness issues commonly encountered by applying scattering alone, how having real space images makes for more confident model selection & fitting constraints.

In the introduction, we have added the following sentence:

“Moreover, even applying scattering techniques, such as liquid-phase X-ray scattering, to characterize UCST materials presents a challenge because appropriate model selection for data fitting is nontrivial and is often guided by TEM data.”

- Issues of whether the small measurement volume of microscopy alone would be representative. Having RSoXS of a larger volume enhances confidence that the characterization is representative of bulk solution.

1b) With regard to RSoXS over conventional SANS, there is of course its advantages in not requiring radiolabeling and its preference for small volumes. These advantages were already heavily discussed and demonstrated in this fine journal in McAfee, et al. Nature Communications 12, 3123 (2021). However, I believe this might be the first example of solution RSoXS in an organic solvent, and its application here is potentially an important demonstration that such an experiment is possible at all considering that the solvent would have a significant cross-section at energies $> \sim 287$ eV. This latter point may be worth mentioning!

We thank the reviewer encouraging us to stress the novelty of our measurement in isopropanol. Indeed, we agree that our study should serve as a demonstration to others that such measurements are even possible. Accordingly, we have added the following:

“It is worth noting that for the energies that we have investigated near the carbon K-edge, the isopropanol solvent itself has a strong cross-section. This consideration has likely prevented others from pursuing liquid RSoXS studies in organic solvents, making our study the first of its kind. However, even without background subtraction, we were still able to obtain sufficient signal intensity to probe the UCST transition, demonstrating the utility of liquid RSoXS to investigate non-aqueous systems in future studies.”

1c) With regard to in-situ TEM, is there anything special beyond beam damage that is learned from the video frame rate capabilities? They are certainly very exciting. “Dynamics” are mentioned a couple of times but there is no quantitative analysis of dynamics. I suppose that the evolution of the structure in both time and temperature is revealed here and there is no necessity to assume quasi-steady state behaviors at different temperatures. Perhaps some of these advantages should be amplified.

We thank the reviewer for their insightful comment. For the videos, we were solely interested in investigating the mechanism and extent of damage by coupling real space LCTEM and *post-mortem* MALDI-IMS. Any analysis of electron beam induced sample dynamics captured in these videos is unrelated to the UCST transition as heating was not applied. As revealed by our damage analysis, a pulsed imaging protocol needs to be used

ensure sample integrity during LCTEM analysis. We then use these optimized imaging conditions to elucidate the UCST transition via pulsed imaging.

1d) Ultimately the key new observation that the authors report: “a UCST-triggered change in morphology from a polydisperse core-shell structure with a complex core to a more-ordered spherical micelle,” [p 13] would seem to this reviewer to be straightforwardly discoverable with conventional approaches so again I urge the authors to provide a more complete explanation on why its discovery is owed to such a novel characterization approach.

We appreciate the reviewer’s thorough comments. As discussed in the introduction, the dynamics we have observed could not be obtained using conventional approaches. As reproduced from the text below, it is impossible to *directly* study UCST transitions using standard microscopy approaches as these approaches are optimized for static materials and aqueous materials. In the absence of direct insight obtained via electron microscopy, applying scattering techniques like SAXS or RSoXS may lead to difficulty in model selection. Thus, for dynamic UCST materials, LCTEM is needed to elucidate both the solvated morphology at room temperature and the thermally triggered dynamics of the sample. For such a complex, multi-block copolymer, leveraging chemically sensitive RSoXS can then corroborate and further refine morphological and mechanistic insight obtained from LCTEM.

“For upper critical solution temperature (UCST) homopolymers, heating above the transition temperature results in an enthalpically driven solubilization, as interactions between polymer chains are weakened in favor of polymer-solvent interactions. In multiblock copolymer amphiphiles, the inclusion of a UCST block can enable temperature triggered nanoscale morphological transitions. The ability of UCST polymers to undergo thermally induced phase transitions holds promise for numerous applications, ranging from insulating materials to catalysis. Despite their utility, there are few studies concerned with probing UCST transitions as these types of phase transformations cannot be imaged by standard microscopy methods that rely on static imaging of dried aliquots. Moreover, the majority of UCST-type polymers do not exhibit thermoresponsiveness under purely aqueous conditions. This limits the utility of cryogenic TEM in deciphering morphological transitions because, despite there being limited examples of extending cryo-TEM to non-aqueous solvents, standard methods are optimized for aqueous samples only. Moreover, even applying scattering techniques, such as liquid-phase X-ray scattering, to characterize UCST materials presents a challenge because appropriate model selection for data fitting is nontrivial and is often guided by TEM data.”

2) RSoXS is a powerful emerging characterization method, and as a community we are just beginning to scratch the surface of its transformative capabilities. I provide comments

here to encourage its rigorous discussion and application in a manner that would better acknowledge conventions and best-practices from solution SAXS and SANS. To this end, I have the following suggestions:

2a) Most SAXS and SANS practitioners would not attempt to fit data that were not rigorously background subtracted. Can that be done with this experimental setup?

We thank the review for their question. For this system, a background subtraction is not possible due to the variability of liquid thickness over separate liquid-cell experiments. While a previous study by McAfee et al. showed that their liquid RSoXS data could be background subtracted, this system used a flow cell to trigger material formation whereas our materials are preformed and thus cannot be flowed in using current liquid RSoXS platforms (McAfee et al. Label-free characterization of organic nanocarriers reveals persistent single molecule cores for hydrocarbon sequestration. *Nat Commun* **12**, 3123 (2021). <https://doi.org/10.1038/s41467-021-23382-8>). To make this clearer, we have added the text below.

“Additionally, the low scattering contrast may partially be attributed to our inability to perform a solvent background subtraction due to the variability of liquid thickness over separate experiments. In future studies, this issue may be ameliorated by creating a flow cell capable of flowing preformed nanomaterials for liquid RSoXS measurements.”

2b) The RSoXS is fit to pure form factor expressions when the TEM indicates a significant structure factor should be present, with agglomerates of spherical particles.

We thank the reviewer for their inquiry. Given the reviewer’s feedback, we have added a structure factor to our fits where possible. See below:

“To further probe the final, trapped morphology after the heating-cooling cycle, we conducted energy dependent RSoXS at 283 eV, 285.2 eV, and 288.8 eV (**Figure 4b**). While the 283 eV data describes the overall micellar morphology, using an energy of 285.2 eV allowed us to enhance contrast from the PS domains. Using a spherical form factor with a sticky hard sphere structure factor, the fit for the 285.2 eV trace revealed a 21 nm PS core radius. As the 283 eV fit shows that the entire core radius was 48 nm, the smaller radius measured at 285.2 eV suggests that, upon cooling, the core likely contains both PS and PEGMeA-co-PEGPhA, pointing to a partial reversibility of the UCST transition.

Using an energy of 288.8 eV results in similar contrast for all three polymer domains, allowing us to probe the overall particle size. Given the similarity in contrast for all three domains, the 288.8 eV trace showed low scattering contrast, and so the overall scattering can be more easily observed by plotting $\ln(I \times Q^2)$ vs $\ln Q$ plot (**Supplementary Figure 15**). Additionally, the low scattering contrast may partially be attributed to our inability to perform a solvent background subtraction due to the variability of liquid thickness over separate experiments. In future studies, this issue may be ameliorated by creating a flow cell capable of flowing preformed nanomaterials for liquid RSoXS measurements. Nonetheless, fitting the 288.8 eV trace to a spherical form factor with a sticky hard sphere structure factor showed an overall sphere diameter of 130 nm. It is worth noting that for the energies that we have investigated near the carbon K-edge, the isopropanol solvent itself has a strong cross-section. This consideration has likely prevented others from pursuing liquid RSoXS studies in organic solvents, making our study the first of its kind. However, even without background subtraction, we were still able to obtain sufficient signal intensity to probe the UCST transition, demonstrating the utility of liquid RSoXS to investigate non-aqueous systems in future studies.”

2c) Is there enough information in the RSoXS pattern to be confident in fitting multiple parameters to the single feature? It could be a Guinier-Porod knee, a structure factor peak, or a form factor maximum. If this were solution SANS, I would insist that the data are significantly overfit.

We thank the review for their comment. We are inclined to agree and have simplified all of our fits to prevent such overfitting. See below:

“Prior to conducting RSoXS measurements, we performed NEXAFS spectroscopy measurements on pure PS, PEGMeA-co-PEGPhA, and PEG films to generate binary contrast functions for the polymers (**Supplementary Figure 13**). At an energy of 283 eV, the contrast between PEGMeA-co-PEGPhA and PEG is small, while the contrast between PS and these two polymers is large. Thus, we used an energy of 283 eV to probe the overall core-shell(-shell) morphology of the PEG-*b*-PEGMeA-co-PEGPhA-*b*-PS polymer structures, as this energy showed the most significant scattering intensity for energies near the carbon K-edge”

“To further probe the final, trapped morphology after the heating-cooling cycle, we conducted energy dependent RSoXS at 283 eV, 285.2 eV, and 288.8 eV (**Figure 4b**). While the 283 eV data describes the overall micellar morphology, using an energy of 285.2 eV allowed us to enhance contrast from the PS domains. Using a spherical form

factor with a sticky hard sphere structure factor, the fit for the 285.2 eV trace revealed a 21 nm PS core radius. As the 283 eV fit shows that the entire core radius was 48 nm, the smaller radius measured at 285.2 eV suggests that, upon cooling, the core likely contains both PS and PEGMeA-co-PEGPhA, pointing to a partial reversibility of the UCST transition.

Using an energy of 288.8 eV results in similar contrast for all three polymer domains, allowing us to probe the overall particle size. Given the similarity in contrast for all three domains, the 288.8 eV trace showed low scattering contrast, and so the overall scattering can be more easily observed by plotting $\ln(I \times Q^2)$ vs $\ln Q$ plot (**Supplementary Figure 15**). Additionally, the low scattering contrast may partially be attributed to our inability to perform a solvent background subtraction due to the variability of liquid thickness over separate experiments. In future studies, this issue may be ameliorated by creating a flow cell capable of flowing preformed nanomaterials for liquid RSoXS measurements. Nonetheless, fitting the 288.8 eV trace to a spherical form factor with a sticky hard sphere structure factor showed an overall sphere diameter of 130 nm. It is worth noting that for the energies that we have investigated near the carbon K-edge, the isopropanol solvent itself has a strong cross-section. This consideration has likely prevented others from pursuing liquid RSoXS studies in organic solvents, making our study the first of its kind. However, even without background subtraction, we were still able to obtain sufficient signal intensity to probe the UCST transition, demonstrating the utility of liquid RSoXS to investigate non-aqueous systems in future studies.”

Supplementary Figure 13. NEXAFS data and binary contrast function. a. Binary contrast functions between different homopolymers. **b.** NEXAFS measurements for homopolymers and UCST triblock copolymer.

2d) Framed differently, there is no uncertainty analysis that supports the fitting, and I am uncomfortable that several features are beyond the diffraction limit of the radiation used. My confidence would be increased were an uncertainty analysis provided showing that the multiparameter fit has low uncertainty.

We thank the reviewer for their comment. See our edits above. Additionally, we have added more detail to the SI on how our fits were conducted and have ensured low fitting errors ($X^2 < 5$) for all measurements.

3) A separate issue that should be addressed is that beam damage is assessed extensively in the TEM experiment (see all the videos), but doesn't appear to be assessed in the RSoXS experiment. In McAfee, et al. Nature Communications 12, 3123 (2021), the authors there assert that a flow cell is used specifically to prevent beam damage effects, and such effects may be significant as soft X-rays are ionizing radiation. The cells used here are not flow cells, they are quiescent. Could the authors please provide evidence that X-ray beam damage does not occur in their system? Perhaps sequential short exposures showing that the pattern is roughly the same for early and late exposures on the same spot would be sufficient.

We thank the review for their comment. While a previous study by McAfee et al. showed that a damage analysis for liquid RSoXS could be done, McAfee used a flow cell for RSoXS experiments to aid in such a beam damage check (McAfee et al. Label-free characterization of organic nanocarriers reveals persistent single molecule cores for hydrocarbon sequestration. *Nat Commun* **12**, 3123 (2021). <https://doi.org/10.1038/s41467-021-23382-8>). In contrast to McAfee's study, our materials are preformed and thus cannot be flowed in using current liquid RSoXS platforms. Moreover, in our study, beam damage in TEM experiments was largely assessed by coupling *post-mortem* MALDI-IMS with LCTEM. These experiments were specifically performed for lower molecularly weight proxies for each block within the UCST polymer. Performing *post-mortem* MALDI-IMS following liquid RSoXS would require a MALDI-IMS instrument to be available onsite at LBNL, which currently is not the case. Moreover, over separate time points, when the morphological transformation appears to be complete, the RSoXS scattering remains constant over separate timepoints, further suggesting the beam damage is limited.

4) If we were to characterize the core-shell assembly that this triblock makes in solution using conventional SANS approaches, we would likely use contrast-variation SANS with

monotonic deuteration of one or more blocks and the extraction of partial scattering functions relevant to single blocks that could be fit separately. This is of course another key advantage of RSoXS in that the SANS experiment would require REALLY extensive deuteration because you might need two different monotonic deuteration schemes to isolate three partial scattering functions. With all of that said, I find the discussion of energy selection in RSoXS and its ability to highlight different materials to be insufficiently quantitative and appearing to suggest incorrect underlying principles of the technique. I am sure the authors didn't intend this; Cheng Wang is one of the pioneers of the technique! However, it should be addressed so that the new technique can put its best foot forward.

Three energies were used: 283 eV, 288.8 eV, and 285.2 eV. 283 eV is described simply as "[the energy that] showed the most significant scattering intensity," 285.2 is said to "probe the PS core," and 288.8 is said to "evaluate the PEG corona." The latter two assertions cite earlier work, and lean on a premise advanced in those earlier works that the selection of energies near key NEXAFS resonances is sufficient to measure a scattering function that is interpretable in ways similar to a partial scattering function from contrast-variation SANS approaches. That is to say, there is an implicit assumption made by the authors that by moving to an energy where one phase in a multiphase sample has significant absorbance, the less-absorbing phases are 1) contrast-matched to each other, and 2) having the same contrast with the highlighted phase. Although this premise was certainly sufficient to convey the promise of the technique in its early days and motivate further investment

in understanding it, it ignores two key factors A) that contrast is developed by both beta (absorbance) and $\Delta\rho$, and B) contrast is a matter of differences, not absolute intensities (or even necessarily peaks) in beta or $\Delta\rho$. Recent work by the Brian Collins group (Ferron et al., Physical Review Letters 119, no. 16 2017: 167801., Figure 1 caption) has demonstrated that a more quantitative framework is required to discuss these physical principles and how they relate to the validity of such assumptions. Ferron et al. describe binary contrast calculations incorporating differences in both beta and $\Delta\rho$, and this framework is more consistent with best-practices in SAXS and SANS, in which contrast is always understood to be based on differences (and in those techniques is due mostly to $\Delta\rho$, not beta!). One way to apply that modern understanding of RSoXS principles to the work under review would be to first calculate binary contrast functions among the 3 blocks from reference dielectric functions in both beta and $\Delta\rho$. There will be $4!/2 = 12$ binary contrast functions, but it depends on whether you consider IPA implicit and also whether you include various degrees of swelling; I would leave that up to the authors. Considering only the 3 "solid"- $\Delta\rho$ contrasts, for 285.2 eV it will be important to show that the PEG vs. PS and PEGMeA-co-PEGPhA vs. PS binary contrasts are of similar magnitude and both much, much larger than the PEG vs. PEGMeA-co-PEGPhA

contrast (because there is an implicit assumption that these are contrast-matched at that energy when solving for the PS core). For 288.8 eV, similarly, it will be important to show that the PEG vs. PEGMeA-co-PEGPhA and PEG vs PS contrasts are similar, and very large relative to the PEGMeA-co-PEGPhA vs. PS contrast. What I have proposed does not require additional experiments (I would cobble together the PEGMeA-co-PEGPhA dielectric function from weighted reference functions in Ade's library <https://www.physics.ncsu.edu/stxm/polymerspectro/> or Gann's <https://github.com/EliotGann/Optical-Constants-Database>), and it is perhaps an hour of additional computation to prove that these assumptions are valid.

We agree with reviewer the quantitative analysis like used in SANS will reveal great structure details with high fidelity. However, the quantitative RSoXS analysis is still in the development phase and the analysis method is evolving with the growing user community. Although limited, there has been some tremendous efforts from several groups that develops quantitative modeling and analysis, most of them are applications in solid thin film samples. In this work, as the reviewer mentioned, we have a quite complicated system involves the organic compound in the organic solvent. This makes the analysis extremely difficult. None the less, we conducted thorough NEXAFS analysis and extracted the complex index of refraction (delta and beta) experimentally on the solid sample of the homopolymers as well as the block copolymers. Admittedly, the optical property and the contrast function of the solid will be different from the solvated form in the organic solution due to factors such as density change, chain stretching and relaxing. This effort is truly a first demonstration that the correlated analysis with multiple in-situ probes that yields greater details.

We appreciate the reviewer's thorough comments. Hoping to address these comments we have added the discussion shown below. Additionally, we have conducted thorough NEXAFS measurements to generate binary contrast functions, as discussed and shown below.

“Given the limitations of SAXS and SANS, we turned here to liquid resonant soft X-ray scattering (RSoXS), where the large scattering cross section of soft X-rays enables analysis of thin transmission samples used in TEM (**Figure 4**). Liquid RSoXS is capable of probing the structure of complex, multicomponent polymeric nanoarchitectures because of its ability to achieve contrast matched scattering through variation of incident X-ray energy. In RSoXS, the scattering intensity is proportional to the energy-dependent contrast function $I \propto |\Delta n(E)|^2$, where Δn denotes the difference between the indices of refraction for two chemical moieties and is parameterized by real and imaginary components via the relation $n(E) = 1 - \delta(E) + i\beta(E)$. The real and imaginary components are typically determined via near-edge X-ray absorption fine structure (NEXAFS)

spectroscopy. Guided by the contrast function, energies near the carbon K-edge can be selected for RSoXS measurements. For example, selecting an energy that allows for contrast matching of the corona block to the solvent enables isolation of scattering contributions from the core. Fitting these scattering data to established models can then provide critical insight into the size and shape of the core. Moreover, by using time-resolved *in situ* RSoXS, we can monitor the morphological development of nanostructures through changes in the form factor. RSoXS data can then be correlated to phenomena directly observed by LCTEM imaging, allowing us to analyze complex phase transformations that would otherwise be challenging to monitor by either RSoXS or LCTEM in isolation. Critically, both LCTEM and liquid RSoXS employ the same liquid-cell holder tip assembly, and thus both experiments subject the polymeric sample to the same inherent confinement effects (**Supplementary Figure 12**).

Prior to conducting RSoXS measurements, we performed NEXAFS spectroscopy measurements on pure PS, PEGMeA-*co*-PEGPhA, and PEG films to generate binary contrast functions for the polymers (**Supplementary Figure 13**). At an energy of 283 eV, the contrast between PEGMeA-*co*-PEGPhA and PEG is small, while the contrast between PS and these two polymers is large. Thus, we used an energy of 283 eV to probe the overall core-shell(-shell) morphology of the PEG-*b*-PEGMeA-*co*-PEGPhA-*b*-PS polymer structures, as this energy showed the most significant scattering intensity for energies near the carbon K-edge (**Figures 4a, Supplementary Figure 14**). Using an energy of 283 eV, at room temperature, a broad trace was observed, indicative of the high polydispersity of the initial assemblies before heating. The room temperature data was fit to a spherical polymer micelle form factor, a morphological model informed by liquid-phase real space imaging. Using this fit revealed that, on average, the initial assemblies had a core radius of 47 nm with a coronal thickness of 48 nm. Upon heating the sample to 60 °C for 10 minutes, an increase in scattering intensity was observed, indicating ordering of the nanoassemblies, in agreement with VT-DLS data showing a tightening of the distribution. Fitting this trace to a polymer micelle form factor showed a core radius of 42 nm and coronal radius of 59 nm. The decrease in the size of the core can be explained by the enhanced solubility of the PEGMeA-*co*-PEGPhA block upon the UCST transition. Presumably, at room temperature, the core is PS and PEGMeA-*co*-PEGPhA-rich, as PEG is the only solvophilic block at room temperature and should thus be found in the outer shell. However, upon heating, PEGMeA-*co*-PEGPhA becomes increasingly solvophilic, leading to an effective increase of the solvophilic corona and a decrease in the insoluble core. This outer corona is “invisible” by LCTEM since this shell is entirely soluble in isopropanol (**Figure 3c**). After being held for 5 additional minutes at 60 °C, we observed a further increase in the scattering intensity. We applied a polymer micelle form factor, yielding a fit with a core radius of 39 nm with coronal thickness of 73 nm, demonstrating further expulsion of PEGMeA-*co*-PEGPhA from the particle core, consistent with TEM imaging data (**Supplementary Figure 10**). By RSoXS, the nanoassemblies remained in

this morphology after being held at 60 °C for 15 more minutes, with a coronal radius of 75 nm and core radius of 38 nm. Upon cooling and holding the sample at < 35 °C for 30 minutes, the sample was fit to a polymer micelle form factor with a core radius of 38 nm with coronal thickness of 65 nm, demonstrating partial reversibility of the UCST transformation given the decrease in coronal thickness.

To further probe the final, trapped morphology after the heating-cooling cycle, we conducted energy dependent RSoXS at 283 eV, 285.2 eV, and 288.8 eV (**Figure 4b**). While the 283 eV data describes the overall micellar morphology, using an energy of 285.2 eV allowed us to enhance contrast from the PS domains. Using a spherical form factor with a sticky hard sphere structure factor, the fit for the 285.2 eV trace revealed a 21 nm PS core radius. As the 283 eV fit shows that the entire core radius was 48 nm, the smaller radius measured at 285.2 eV suggests that, upon cooling, the core likely contains both PS and PEGMeA-co-PEGPhA, pointing to a partial reversibility of the UCST transition.

Using an energy of 288.8 eV results in similar contrast for all three polymer domains, allowing us to probe the overall particle size.^{24, 25} Given the similarity in contrast for all three domains, the 288.8 eV trace showed low scattering contrast, and so the overall scattering can be more easily observed by plotting $\ln(I \times Q^2)$ vs $\ln Q$ plot (**Supplementary Figure 15**). Additionally, the low scattering contrast may partially be attributed to our inability to perform a solvent background subtraction due to the variability of liquid thickness over separate experiments. In future studies, this issue may be ameliorated by creating a flow cell capable of flowing preformed nanomaterials for liquid RSoXS measurements. Nonetheless, fitting the 288.8 eV trace to a spherical form factor with a sticky hard sphere structure factor showed an overall sphere diameter of 130 nm. It is worth noting that for the energies that we have investigated near the carbon K-edge, the isopropanol solvent itself has a strong cross-section. This consideration has likely prevented others from pursuing liquid RSoXS studies in organic solvents, making our study the first of its kind. However, even without background subtraction, we were still able to obtain sufficient signal intensity to probe the UCST transition, demonstrating the utility of liquid RSoXS to investigate non-aqueous systems in future studies.”

Supplementary Figure 13. NEXAFS data and binary contrast function. a. Binary contrast functions between different homopolymers. **b.** NEXAFS measurements for homopolymers and UCST triblock copolymer.

Modifying the RSoXS discussion and making its assumptions regarding energy selection more quantitative would put the RSoXS discussion on firmer footing, dispel questions about how the technique might compare to contrast-variation SANS, and potentially place an even stronger spotlight on the exciting new capabilities this technique brings to soft matter structural characterization.

Reviewers' Comments:

Reviewer #1:

Remarks to the Author:

In their revised manuscript, Korpanty et al improve their work significantly, but there are still some missing pieces as described below before I'm comfortable recommending publication.

First, I thank the authors for the clarification of the damage reduction experiment and their added evidence that existing damage isn't the source of the nanoparticle shrinkage. However, the statements on pg6 are too strong: "allowing us to establish a safe imaging window...we ensure that under low flux conditions, the observed particle behavior is the result of the UCST transition and not the result of electron beam damage." I'm not aware of any damage thresholds where >40% mass loss is considered "safe" and able to "ensure" no artifacts from damage. The added evidence later helps to support this interpretation, but that evidence has not yet been presented and so their statements here imply such a threshold. Additionally, it would be prudent to incorporate in the main text the reasoning behind the choice of 0.33 e/sA^2 as the "low flux condition" rather than say a lower flux that has no measurable difference from the unimaged control.

The agreement from the authors that "the polymer chemical components cannot be measured/determine using the false color filter" is appreciated, but the modifications in the main text do not reflect this. On pg8, the authors still explicitly label colors directly with chemical components contradicting their statements in the rebuttal. The authors need to actually change the main text discussion to reflect that "the false color filter is a sensitive method for looking at the density" as discussed in the rebuttal and not explicitly label colors with chemical components.

The authors have significantly improved the RSoXS analysis section, but several issues remain.

1) First my original requests need to be fulfilled where I originally wrote "the precise fit function needs to be explicitly described or referenced somewhere". For example, the described "spherical form factor" used doesn't have a corona term nor "penetration factor". The function used, needs to be described sufficiently such that it is clear how each parameter is incorporated into the model. Also the parameters need to be presented with units.

2) The authors on pg10 write "an increase in scattering intensity was observed" this is not discernable in the current figure 4a (presumably should be seen between the two bottom traces). Please plot this in a way that demonstrates this observation more clearly.

3) The provided contrast functions are much appreciated. Can the authors highlight the three energies they used somehow on these plots?

4) Probably most importantly, I really don't see how the 285.2eV and 288.8eV fits are extracting much of anything, and so I don't think these fits and related discussion is accurate. In SI figure 15 the authors show q-square scaled data but not fits. Perhaps there are ways like this to point to where the data reveal these critical features and how the fits pick this up. Otherwise quoting and interpreting fit values (e.g. the proposed 21 nm PS core radius or the 130 nm overall sphere diameter) should be removed.

If the authors can resolve these remaining issues, then I would be comfortable with recommending publication.

Reviewer #2:

Remarks to the Author:

The authors have addressed many of my comments and those of the other reviewer in this submission. The RSoXS analysis has been dramatically revised. I encourage the editors to continue consideration for publication in this fine journal provided the following concerns are addressed.

- The authors now provide quantitative binary contrast calculations to support their energy selection, and I'll further note that their interpretation of which parts of the morphology are highlighted at different energies has changed, so it certainly seems to have been a useful exercise. I would strongly encourage the authors to put figure S13 in the main text and refer to it as they discuss energy selection. The emerging nature of the technique and the fact that both reviewers highlighted

quantitative rationale for energy selection suggests that these are not details to be hidden in the SI. I would encourage a logarithmic Figure S13a axis.

- It is good that the authors have settled on a simpler model for fitting their RSoXS, but there is insufficient information here for anyone to reproduce the fits; as far as I can tell there is no citation or description of the analytical fitting equation(s) that was used, nor the optimization method or software that was used to fit the data. Further, some discussion about the reconciliation of these fits with real optical parameters is encouraged, particularly around the following points: 1) the SLDs in the fit are real only; complex numbers are required to convey beta, and 2) the SLDs vary in sign and magnitude significantly with both phase (corona, core, medium) and energy. I realize the authors are using a more conventional SAXS /SANS fitting approach in a kind of exploratory way to extract spatial parameters, but I would encourage them to at least contextualize their SLD results, particularly with respect to the question of whether the relative ranking of delta-SLD in the fits is consistent with the trends shown in Figure S13, which it should be.

- The authors indicate that they are unable to assess RSoXS sample damage. This is unfortunate, as their own work and that of McAfee et al. suggest that there will be some sample damage and potentially alteration of the scattering pattern. The authors did not address my question of whether simple short, sequential exposures of their quiescent cells would prove that there is not substantial evolution of the material in the beam. I do not agree that flow cells are required to assess beam damage.

Reviewer #1 (Remarks to the Author):

In their revised manuscript, Korpanty et al improve their work significantly, but there are still some missing pieces as described below before I'm comfortable recommending publication.

First, I thank the authors for the clarification of the damage reduction experiment and their added evidence that existing damage isn't the source of the nanoparticle shrinkage. However, the statements on pg6 are too strong: "allowing us to establish a safe imaging window...we ensure that under low flux conditions, the observed particle behavior is the result of the UCST transition and not the result of electron beam damage." I'm not aware of any damage thresholds where >40% mass loss is considered "safe" and able to "ensure" no artifacts from damage. The added evidence later helps to support this interpretation, but that evidence has not yet been presented and so their statements here imply such a threshold. Additionally, it would be prudent to incorporate in the main text the reasoning behind the choice of $0.33 \text{ e}^-/\text{s}\text{\AA}^2$ as the "low flux condition" rather than say a lower flux that has no measurable difference from the unimaged control.

We thank the reviewer for pointing out this error. We agree that we need to modify these statements and have done so, as reproduced below:

"On the other hand, low flux conditions showed enhanced mass signal retention, allowing us to limit the degradation of the examined polymers. Thus, our results suggest that under low flux conditions, the observed particle behavior is likely the result of the UCST transition and not the result of electron beam damage."

While we can limit the damaging effect of the electron beam, it is impossible to eliminate damage entirely. Given the balance between having sufficient signal-to-noise and limiting sample damage, it is likely impossible to obtain data where the imaged sample shows signal retention identical to the unimaged control. On this we agree with the reviewer that we need to justify our flux selection and have added the following:

"The low flux condition of $0.33 \text{ e}^- \text{\AA}^{-2} \text{ s}^{-1}$ was chosen because it enabled visibility of nanostructures without clear structure degradation during imaging."

The agreement from the authors that "the polymer chemical components cannot be measured/determined using the false color filter" is appreciated, but the modifications in the main text do not reflect this. On pg8, the authors still explicitly label colors directly with chemical components contradicting their statements in the rebuttal. The authors need to actually change the main text discussion to reflect that "the false color filter is a sensitive

method for looking at the density” as discussed in the rebuttal and not explicitly label colors with chemical components.

We agree and have erased all statements where colors in the false color image are correlated with chemical components. We now amend the sentences referring to the false color image as follows:

“This image processing procedure allows us to probe subtle differences in density and highlights that initially...”

The authors have significantly improved the RSoXS analysis section, but several issues remain.

1) First my original requests need to be fulfilled where I originally wrote “the precise fit function needs to be explicitly described or referenced somewhere”. For example, the described “spherical form factor” used doesn’t have a corona term nor “penetration factor”. The function used, needs to be described sufficiently such that it is clear how each parameter is incorporated into the model. Also the parameters need to be presented with units.

We thank the reviewer for their comments. We have added further detail on the models used to obtain fits, as shown below, and have added units to all parameters. Additionally, the information below has now been included in the SI:

The 1D scattering intensity for the polymer micelle model is calculated according to the equations given by Pedersen. In the following discussion, we reproduce the description for the polymer micelle model compiled by SASView. The micelle core of radius r is represented by N polymer heads, each of volume V_{core} . Gaussian random coil tails, with a radius of gyration R_g , are distributed around the spherical core. These coils are centered at a distance of $r + d \cdot R_g$ from the micelle center. Here, d is a penetration term on the order unity. For each coil, a volume of V_{corona} can be defined.

$$P(q) = N^2\beta_s^2\Phi(qr)^2 + N\beta_c^2P_c(q) + 2N^2\beta_s\beta_cS_{sc}(q) + N(N-1)\beta_c^2S_{cc}(q)$$

$$\beta_s = V_{core}(\rho_{core} - \rho_{solvent})$$

$$\beta_c = V_{corona}(\rho_{corona} - \rho_{solvent})$$

where ρ_{corona} , $\rho_{solvent}$, ρ_{core} are the scattering length densities. For the spherical core of radius r

$$\Phi(qr) = \frac{\sin(qr) - qr\cos(qr)}{(qr)^3}$$

And for the Gaussian coils

$$P_c(q) = 2[\exp(-Z) + Z - 1] / Z^2$$

$$Z = (qR_g)^2$$

The core-to-corona and corona-to-corona cross terms are approximated by:

$$S_{sc}(q) = \frac{\Phi(qr)\psi(Z)\sin(q(r + d \cdot R_g))}{q(r + d \cdot R_g)}$$

$$S_{cc}(q) = \left[\frac{\sin(q(r + d \cdot R_g))}{q(r + d \cdot R_g)} \right]^2$$

$$\psi(Z) = \frac{1 - \exp^{-Z}}{Z}$$

For the sticky hard sphere model, the perturbation parameter, τ , should be fixed between 0.01 and 0.1. The stickiness, ϵ , is used to adjust the interaction strength. The stickiness is a function of both the perturbation parameter and the interaction strength. ϵ and τ are defined in terms of the hard sphere diameter ($\sigma = 2R$), the width of the square well, Δ , and the depth of the well, U_0 .

$$\epsilon = (1/12\tau) \exp(u_0 / kT)$$

$$T = \Delta / (\sigma + \Delta)$$

where the interaction potential is

$$U(r) = \begin{cases} \infty & r < \sigma \\ -U_0 & \sigma \leq r \leq \sigma + \Delta \\ 0 & r > \sigma + \Delta \end{cases}$$

2) The authors on pg10 write “an increase in scattering intensity was observed” this is not discernable in the current figure 4a (presumably should be seen between the two bottom traces). Please plot this in a way that demonstrates this observation more clearly.

We have adjusted this statement to “a scattering feature was observed at $q \sim 0.007 \text{ nm}^{-1}$.”

3) The provided contrast functions are much appreciated. Can the authors highlight the three energies they used somehow on these plots?

We appreciate the reviewer's advice and have adjusted the figure to make our selected energies clearer. We also moved this figure to the main text at the advice of reviewer #2.

4) Probably most importantly, I really don't see how the 285.2eV and 288.8eV fits are extracting much of anything, and so I don't think these fits and related discussion is accurate. In SI figure 15 the authors show q-square scaled data but not fits. Perhaps there are ways like this to point to where the data reveal these critical features and how the fits pick this up. Otherwise quoting and interpreting fit values (e.g. the proposed 21 nm PS core radius or the 130 nm overall sphere diameter) should be removed. If the authors can resolve these remaining issues, then I would be comfortable with recommending publication.

We thank the reviewer for their feedback. The original Supplementary Figure 15 was meant to highlight that the trace measured at 285.2 eV does indeed have structural features that are overwhelmed by the background signal. NOTE: This figure is now numbered, Supplementary Figure 14 in the revised SI.

We acknowledge that the models are necessarily simplified, and the feature sizes we extract from our fits are not precise. We have adjusted our interpretation in the text to give statements making it clear that our model is a simplified view of the system and have done so. Specifically, rather than using terms like "the fit revealed," we now write "the fit suggested" in all such instances. Additionally, we added the statement below.

"Due to our use of simplified models, we stress the importance of the ability of RSoXS to probe the dynamic behavior of the sample and relative change in size upon heating, rather than the precise size of scattering features."

Reviewer #2 (Remarks to the Author):

The authors have addressed many of my comments and those of the other reviewer in this submission. The RSoXS analysis has been dramatically revised. I encourage the editors to continue consideration for publication in this fine journal provided the following concerns are addressed.

- The authors now provide quantitative binary contrast calculations to support their energy selection, and I'll further note that their interpretation of which parts of the morphology are highlighted at different energies has changed, so it certainly seems to have been a useful exercise. I would strongly encourage the authors to put figure S13 in the main text and refer to it as they discuss energy selection. The emerging nature of the technique and the fact that both reviewers highlighted quantitative rationale for energy selection suggests that these are not details to be hidden in the SI. I would encourage a logarithmic Figure S13a axis.

We thank the reviewer for their comments. We have moved this figure to the main text and have plotted it on a log scale.

- It is good that the authors have settled on a simpler model for fitting their RSoXS, but there is insufficient information here for anyone to reproduce the fits; as far as I can tell there is no citation or description of the analytical fitting equation(s) that was used, nor the optimization method or software that was used to fit the data. Further, some discussion about the reconciliation of these fits with real optical parameters is encouraged, particularly around the following points:

In light of the reviewer's comments, we have added the following to the supplementary information as also requested by reviewer #1:

The 1D scattering intensity for the polymer micelle model is calculated according to the equations given by Pedersen. In the following discussion, we reproduce the description for the polymer micelle model compiled by SASView. The micelle core of radius r is represented by N polymer heads, each of volume V_{core} . Gaussian random coil tails, with a radius of gyration R_g , are distributed around the spherical core. These coils are centered at a distance of $r + d \cdot R_g$ from the micelle center. Here, d is a penetration term on the order unity. For each coil, a volume of V_{corona} can be defined.

$$P(q) = N^2 \beta_s^2 \Phi(qr)^2 + N \beta_c^2 P_c(q) + 2N^2 \beta_s \beta_c S_{sc}(q) + N(N-1) \beta_c^2 S_{cc}(q)$$

$$\beta_s = V_{core}(\rho_{core} - \rho_{solvent})$$

$$\beta_c = V_{corona}(\rho_{corona} - \rho_{solvent})$$

where ρ_{corona} , ρ_{solvent} , ρ_{core} are the scattering length densities. For the spherical core of radius r

$$\Phi(qr) = \frac{\sin(qr) - qr\cos(qr)}{(qr)^3}$$

And for the Gaussian coils

$$P_c(q) = 2[\exp(-Z) + Z - 1] / Z^2$$

$$Z = (qR_g)^2$$

The core-to-corona and corona-to-corona cross terms are approximated by:

$$S_{sc}(q) = \frac{\Phi(qr)\psi(Z)\sin(q(r + d \cdot R_g))}{q(r + d \cdot R_g)}$$

$$S_{cc}(q) = \left[\frac{\sin(q(r + d \cdot R_g))}{q(r + d \cdot R_g)} \right]^2$$

$$\psi(Z) = \frac{1 - \exp^{-Z}}{Z}$$

For the sticky hard sphere model, the perturbation parameter, τ , should be fixed between 0.01 and 0.1. The stickiness, ϵ , is used to adjust the interaction strength. The stickiness is a function of both the perturbation parameter and the interaction strength. ϵ and τ are defined in terms of the hard sphere diameter ($\sigma = 2R$), the width of the square well, Δ , and the depth of the well, U_0 .

$$\epsilon = (1/12\tau) \exp(u_0 / kT)$$

$$\tau = \Delta / (\sigma + \Delta)$$

where the interaction potential is

$$U(r) = \begin{cases} \infty & r < \sigma \\ -U_0 & \sigma \leq r \leq \sigma + \Delta \\ 0 & r > \sigma + \Delta \end{cases}$$

- 1) the SLDs in the fit are real only; complex numbers are required to convey beta, and
- 2) the SLDs vary in sign and magnitude significantly with both phase (corona, core, medium) and energy. I realize the authors are using a more conventional SAXS /SANS

fitting approach in a kind of exploratory way to extract spatial parameters, but I would encourage them to at least contextualize their SLD results, particularly with respect to the question of whether the relative ranking of delta-SLD in the fits is consistent with the trends shown in Figure S13, which it should be.

We agree with the reviewer that additional information on beta and delta would be beneficial. Accordingly, we have moved our contrast function figure to the main text. Here, the contrast is based on the difference between delta and beta. While SASView allows us to calculate both the real and imaginary components of the SLD, only the real part can be used in our fitting. In other words, we cannot contextualize the SLD without considering both delta and beta, but here our model only uses delta. We acknowledge this is an oversimplification and we plan to address this in future work. Here, our main goal was to leverage TEM and RSoXS in tandem to study a complex system, and in future work, we will focus on detailed RSoXS studies.

- The authors indicate that they are unable to assess RSoXS sample damage. This is unfortunate, as their own work and that of McAfee et al. suggest that there will be some sample damage and potentially alteration of the scattering pattern. The authors did not address my question of whether simple short, sequential exposures of their quiescent cells would prove that there is not substantial evolution of the material in the beam. I do not agree that flow cells are required to assess beam damage.

We agree that damage is an important consideration. Accordingly, below we show two subsequent exposures at 10 s each taken for the sample at 283 eV at the end of the heating cooling cycle. In both traces, you can see the scattering feature remains constant with no visible signs of damage. These data suggest that the dynamics of the sample are not largely the result of beam damage and rather the result of the thermal stimulus. We have added this figure to the supplementary information and hope the reviewer and readers find it satisfactory evidence that beam exposure does not manifest in visible RSoXS artifacts.

Supplementary Figure 15. Two RSoXS traces measured for a solution of PEG-*b*-PEGMeA-co-PEGPhA and PEGMeA in IPA at 283 eV after the heating and cooling cycle. This data highlights that upon subsequent exposures, the sample does not undergo visible beam induced degradation.

Reviewers' Comments:

Reviewer #1:

Remarks to the Author:

With their further revisions, the authors have improved their manuscript significantly. I can recommend this work for publication if the following relatively simple changes can be made.

- 1) The revisions on page 6 help, but need further fixing. The last new sentence in the paragraph "Thus, our results suggest...the observed particle behavior is likely the result of the UCST transition..." is out of place. The authors have not described the observed behavior yet and merely established that 'only' 40% mass loss has occurred under their chosen imaging conditions. This sentence is better placed at the end of the next paragraph where the behavior in question is described along with further evidence that the behavior is not the result of degradation. Thus please move/alter this sentence to be at the end of the next paragraph. Also the second word "safe" in this paragraph is too strong – replace with something relative like "gentler" or "less damaging". Finally, the "stroboscopic" measurement is presumably an important aspect to limiting damage but is first introduced well after this imaging method has been discussed at length. Please mention/discuss it earlier.
- 2) Figure 4a and 4b: "POEGA" is presumably the thermo-responsive block, but its definition is never mentioned anywhere. Please identify this more directly somehow. Also, the contrast functions (4a) are changing by orders of magnitude in the yellow highlighted regions making it impossible to know what the relative contrasts are. Please narrow the energy range of the contrast functions in the graph so the values of the contrast are better defined within the yellow regions (e.g. only show from 275-295eV or so). Finally, IPA likely has a complicated index of refraction here, which is a challenge to measure. This missing optical parameter and set of contrast functions should be mentioned somewhere. (I'm not asking for these to be measured/incorporated into the analysis. Just make it clear that this is an important factor that is not addressed here.)
- 3) The added detail for the RSoXS model fits are appreciated as are the fixed parameter values. Can the authors confirm that the only open parameters in the fit are the core and shell radii/thicknesses? (If there are other open parameters, they should be provided.) Can the authors also indicate in the paper whether polydispersity is modeled? (It looks like it's not, but it's important for reproducibility to indicate this.) It would help the reader of the paragraph on page 10 to be consistent with always listing the core radius before the corona thickness. (One is listed backwards.) Finally, it might be good to list these five fit parameter sets in a table somewhere just for ease of readability instead of hunting around the paragraph for them.

Reviewer #2:

Remarks to the Author:

I thank the authors for addressing my comments and those of the other reviewer. My concerns have been satisfied with this round of review. The other reviewer brings up excellent points that should be addressed. If the other reviewer is also in agreement, I would encourage the editors to accept this manuscript for publication.

Reviewer #1 (Remarks to the Author):

With their further revisions, the authors have improved their manuscript significantly. I can recommend this work for publication if the following relatively simple changes can be made.

1) The revisions on page 6 help, but need further fixing. The last new sentence in the paragraph “Thus, our results suggest...the observed particle behavior is likely the result of the UCST transition...” is out of place. The authors have not described the observed behavior yet and merely established that ‘only’ 40% mass loss has occurred under their chosen imaging conditions. This sentence is better placed at the end of the next paragraph where the behavior in question is described along with further evidence that the behavior is not the result of degradation. Thus please move/alter this sentence to be at the end of the next paragraph. Also the second word “safe” in this paragraph is too strong – replace with something relative like “gentler” or “less damaging”. Finally, the “stroboscopic” measurement is presumably an important aspect to limiting damage but is first introduced well after this imaging method has been discussed at length. Please mention/discuss it earlier.

We thank the review for bringing this to our attention. We have moved this sentence to the end of the next paragraph where we discuss the low flux experiment. We have also changed the word “safe” to “gentle.” Finally, before introducing our use of stroboscopic imaging, we added the following sentence:

“Here, we employed stroboscopic rather than continuous imaging conditions to limit the extent of electron beam induced damage to the polymeric materials.”

2) Figure 4a and 4b: “POEGA” is presumably the thermo-responsive block, but its definition is never mentioned anywhere. Please identify this more directly somehow. Also, the contrast functions (4a) are changing by orders of magnitude in the yellow highlighted regions making it impossible to know what the relative contrasts are. Please narrow the energy range of the contrast functions in the graph so the values of the contrast are better defined within the yellow regions (e.g. only show from 275-295eV or so). Finally, IPA likely has a complicated index of refraction here, which is a challenge to measure. This missing optical parameter and set of contrast functions should be mentioned somewhere. (I’m not asking for these to be measured/incorporated into the analysis. Just make it clear that this is an important factor that is not addressed here.)

We thank the reviewer for pointing out this error. We have added the statement “...where POEGA is an abbreviation for the oligoethylene glycol copolymer PEGMeA-co-PEGPhA” to the figure caption. We also have altered Figure 4 as suggested to highlight the region 275-295 eV. Finally, at the advice of the reviewer, we have added the following statement:

“We also note that IPA likely has a complicated index of refraction, which is challenging to measure and so in our analysis, we are missing an optical parameter and set of contrast functions associated with the solvent.”

3) The added detail for the RSoXS model fits are appreciated as are the fixed parameter values. Can the authors confirm that the only open parameters in the fit are the core and shell radii/thicknesses? (If there are other open parameters, they should be provided.) Can the authors also indicate in the paper whether polydispersity is modeled? (It looks like it's not, but it's important for reproducibility to indicate this.) It would help the reader of the paragraph on page 10 to be consistent with always listing the core radius before the corona thickness. (One is listed backwards.) Finally, it might be good to list these five fit parameter sets in a table somewhere just for ease of readability instead of hunting around the paragraph for them.

We appreciate the reviewer's approval of our edits. We did model the effect of the polydispersity for the 288.8 eV measurement (as the polymer micelle model used for the 283 eV traces does not allow polydispersity to be varied, and the 285.2 eV trace did not need polydispersity to be varied in order to obtain a sufficient fit). We thank the reviewer for noting the importance of indicating this, and we have added the polydispersity to the supplementary information document. We have also ensured that all mentions of core and corona sizes are listed in a consistent order for clarity. Finally, we have added a table to the supplementary information listing the measured parameters, as shown below.

Supplementary Table 4. Core radii and radii of gyration (R_g) for coronal chains measured at 283 eV as a function of temperature and time.

Measurement	Core radius	R_g of coronal chains
25 °C, 0 min	47 nm	48 nm
60 °C, 10 min	42 nm	59 nm
60 °C, 15 min	39 nm	73 nm
60 °C, 30 min	38 nm	75 nm
< 35 °C, 30 min	38 nm	65 nm

Reviewer #2 (Remarks to the Author):

I thank the authors for addressing my comments and those of the other reviewer. My concerns have been satisfied with this round of review. The other reviewer brings up excellent points that should be addressed. If the other reviewer is also in agreement, I would encourage the editors to accept this manuscript for publication.

We thank the reviewer for their kind remarks.